# Influence of Functional Variations in Genes of Neurotrophins and Neurotransmitter Systems on the Development of Retinopathy of Prematurity

**DOI:** 10.3390/ijms26030898

**Published:** 2025-01-22

**Authors:** Mariza Fevereiro-Martins, Ana Carolina Santos, Carlos Marques-Neves, Hercília Guimarães, Manuel Bicho

**Affiliations:** 1Faculdade de Medicina, Universidade de Lisboa, Av. Professor Egas Moniz, 1649-028 Lisboa, Portugal; carolinasantos@medicina.ulisboa.pt (A.C.S.); cmneves@medicina.ulisboa.pt (C.M.-N.); manuelbicho@medicina.ulisboa.pt (M.B.); 2Grupo Ecogenética e Saúde Humana, Instituto de Saúde Ambiental-ISAMB, Laboratório Associado Terra, Faculdade de Medicina, Universidade de Lisboa, Av. Professor Egas Moniz, 1649-028 Lisboa, Portugal; 3Instituto de Investigação Científica Bento da Rocha Cabral, Calçada Bento da Rocha Cabral 14, 1250-012 Lisboa, Portugal; 4Departamento de Oftalmologia, Hospital Cuf Descobertas, Rua Mário Botas, 1998-018 Lisboa, Portugal; 5Centro de Estudos das Ciências da Visão, Faculdade de Medicina, Universidade de Lisboa, Av. Professor Egas Moniz, Piso 1C, 1649-028 Lisboa, Portugal; 6Departamento de Ginecologia-Obstetrícia e Pediatria, Faculdade de Medicina, Universidade do Porto, Alameda Prof. Hernâni Monteiro, 4200-319 Porto, Portugal; hguimara@med.up.pt

**Keywords:** retinopathy of prematurity, nerve growth factor, brain-derived neurotrophic factor, tyrosine hydroxylase, erythropoietin, neurotrophic factors, retinal vascular disease, oxidative stress, gene expression, angiogenesis

## Abstract

Retinal neurodevelopment, vascularization, homeostasis, and stress response are influenced by factors such as *nerve growth factor* (*NGF*), *brain-derived neurotrophic factor* (*BDNF*), *tyrosine hydroxylase* (*TH*), and *erythropoietin* (*EPO*). As retinopathy of prematurity (ROP) is a neurovascular retinal disease, this study analyzed the contributions of *NGF* (rs6330), *BDNF* (rs7934165), *TH* (rs10770141), and *EPO* (rs507392) genetic functional polymorphisms to the modulation of hematological and biochemical parameters of the first week of life and their association with ROP development. A multicenter cohort of 396 preterm infants (gestational age < 32 weeks or birth weight < 1500 g) was genotyped using MicroChip DNA and iPlex MassARRAY^®^ platform. Multivariate regression followed univariate assessment of ROP risk factors. *NGF* (GG) genotype was associated with a higher ROP risk (OR = 1.79), which increased further (OR = 2.38) when epistatic interactions with *TH* (allele C) and *BDNF* (allele G) were present. Significant circulating biomarker differences, including bilirubin, erythrocytes, monocytes, neutrophils, lymphocytes, and platelet markers, were found between ROP and non-ROP groups, with variations depending on the polymorphism. These findings suggest that *NGF* (rs6330) and its interactions with related genes contribute to ROP risk, providing valuable insights into the genetic and biological mechanisms underlying the disease and identifying potential predictive biomarkers.

## 1. Introduction

Retinopathy of prematurity (ROP) is a complex retinal neurovascular condition and one of the leading causes of childhood blindness, especially affecting preterm infants. The incidence of ROP is inversely correlated with gestational age (GA) and birth weight (BW), with the highest risk seen in infants born before 31 weeks or weighing less than 1250 grams [1]. Globally, ROP significantly contributes to childhood blindness, with its prevalence varying according to access to neonatal care and screening protocols [2]. As part of the central nervous system, the retina exhibits a similar susceptibility to damage resulting from preterm birth. This susceptibility, coupled with complex neurovascular interactions, often results in long-term neurodevelopmental and visual impairments [3]. ROP progresses through two distinct phases: an initial phase of arrested retinal vascular development and microvascular degeneration, often triggered by high oxygen exposure, followed by a second phase characterized by abnormal neovascularization in response to hypoxia from early vascular regression [4]. The retina’s neural network, including neurons, Müller cells, astrocytes, and endothelial cells, is pivotal in maintaining retinal function, vascular homeostasis, and repair. This neurovascular interaction is crucial for normal vascular development and is a key driver in the onset of ROP [5].

Maternal stress, driven by genetic and environmental factors, is increasingly recognized as a contributor to preterm birth [6]. Stress triggers the activation of the hypothalamic-pituitary-adrenal (HPA) and sympathetic-adreno-medullary system, leading to elevated levels of stress hormones critical for fetal development [7]. Postnatal stress experienced in the neonatal intensive care unit may further impair neurodevelopment in preterm infants, potentially exacerbating conditions such as ROP [8,9].

Neurotrophic factors, such as nerve growth factor (NGF) and brain-derived neurotrophic factor (BDNF), are crucial for the differentiation, maturation, and survival of neurons in both the central and peripheral nervous systems [10]. These factors, alongside glucocorticoids, are essential for maintaining neuronal connectivity, with BDNF-mediated neuroplasticity being particularly sensitive to stress-induced glucocorticoid responses [11]. NGF, the prototype neurotrophic factor, was discovered in the 1950s, with BDNF identified three decades later [10]. Both factors exhibit significant fluctuations in brain and circulation levels following stress exposure in both animal models and humans [12].

Tyrosine hydroxylase (TH), the rate-limiting enzyme in catecholamine synthesis within the adrenal medulla and dopaminergic and noradrenergic neurons, plays a pivotal role in stress responses and metabolic regulation [13,14]. Catecholamines are essential neurotransmitters that regulate homeostasis under both stress and basal conditions [15]. Additionally, TH is crucial for thermogenesis in brown adipose tissue (BAT), a metabolically active organ characterized by dense sympathetic innervation and a high mitochondrial content [15,16]. BAT relies on uncoupling protein 1 (UCP1) to generate heat through oxidative phosphorylation, a process requiring significant oxygen consumption. Disruptions in BAT activity could influence systemic oxygen availability, potentially affecting retinal vascularization. In preterm infants, who have limited BAT reserves and immature thermoregulatory mechanisms [16], these factors may have implications for the development and progression of ROP.

BDNF crosses the blood–brain barrier and is modulated by various stressors, playing a crucial role in the survival and maturation of retinal ganglion cells [10,17]. Through its interaction with the serotonergic system, one of the oldest neurotransmitter systems, BDNF regulates the HPA axis and catecholamine effects, influencing brain development, neuroplasticity, and BDNF expression [10,18].

Erythropoietin (EPO), traditionally recognized for its role in erythropoiesis, also exhibits neuroprotective properties in the central nervous system and contributes to metabolic regulation. Its influence on adrenocorticotropic hormone (ACTH) levels highlights its broader involvement in metabolic and stress responses through the HPA axis [19].

Stress-induced hormonal responses play a critical role in metabolic regulation, with hyperglycemia being a common consequence of the HPA axis activation in preterm infants. This condition, affecting up to 80% of very-low-birth-weight neonates, arises from stress-related hormones, limited insulin production, and increased insulin resistance, compounded by delayed enteral feeding, intravenous glucose administration, and the use of inotropes and corticosteroids [20,21]. While some studies suggest that hyperglycemia exacerbates oxidative stress and systemic inflammation, potentially increasing the risk or severity of ROP [20,22], others report conflicting findings [23]. Regardless, the interplay between hyperglycemia, oxidative stress, and inflammatory pathways highlights its potential impact on retinal neurovascular regulation in the context of ROP.

Considering the roles of NGF, BDNF, TH, and EPO in retinal neurodevelopment, as well as vascularization [24,25,26,27,28], homeostasis, and stress regulation [18,19,29,30], we hypothesize that functional polymorphisms in these genes may contribute to the risk of ROP development. Thus, this study aims to investigate the contributions of *NGF* (rs6330), *TH* (rs10770141), *BDNF* (rs7934165), and *EPO* (rs507392) polymorphisms to hematological and biochemical parameters in the first week of life and assess their association with ROP development.

## 2. Results

### 2.1. Clinical Characteristics

This study included 396 preterm infants, of which 238 (60.1%) did not develop ROP and 158 (39.9%) were diagnosed with ROP. Table 1 summarizes the clinical characteristics of these infants, categorized by the absence or presence of ROP.

GA and BW were significantly lower in the ROP group (*p* < 0.001), while sex distribution was similar. Among ROP cases, 15.6% had an Apgar score below seven at five minutes. Infants with ROP required more days of mechanical ventilation, had more frequent hyperglycemia, and received more red blood cell (RBC) and platelet transfusions (*p* < 0.001). Hematological differences in the ROP group included lower median values of erythrocytes, hemoglobin, lymphocytes, and platelets, with higher levels of red blood cell distribution width (RDW), erythroblast count, neutrophil/lymphocyte ratio (NLR), basophils, and direct bilirubin.

### 2.2. BDNF, NGF, EPO, and TH Polymorphisms

The genotype GG of the ***NGF*** (rs6330) polymorphism presented a 1.79-fold risk of developing ROP (Table 2). No other significant differences were found between the distributions of different alleles or genotypes and the development of ROP.

### 2.3. NGF Polymorphism, Hematological and Biochemical Parameters, and Incidence of ROP

Higher values of lymphocyte/monocyte ratio (LMR) were found for carriers of the genotype AA and homozygotes (AA + GG) of ***NGF*** only among premature infants who did not develop ROP (Table 3). Additionally, in this group, the median C-reactive protein (CRP) level was significantly lower for the allele G of ***NGF***, and this difference remained significant after adjustment for the number of RBC transfusions and GA, risk factors determined by multivariate regression analysis. In the group that developed ROP, CRP levels were higher for the allele G, although this difference was not statistically significant.

For total bilirubin, a higher median value was observed in carriers of the ***NGF*** allele G who did not develop ROP, and a lower median value was observed in carriers of the same allele who developed ROP. The difference was statistically significant after adjustment only for the group that developed ROP.

### 2.4. TH Polymorphism, Hematological and Biochemical Parameters, and Incidence of ROP

The median mean corpuscular hemoglobin (MCH) values were significantly and independently lower for the genotype CC compared to the allele T of the ***TH*** polymorphism only in the group of infants who did not develop ROP (Table 4). Additionally, the genotype CC in this group presented higher median leukocyte counts.

In the group of infants who developed ROP, homozygotes for the *TH* polymorphism (CC and TT) had significantly and independently higher neutrophil percentages and NLR, but lower lymphocyte percentages compared to heterozygotes. In the same group, carriers of the allele C of the ***TH*** polymorphism exhibited significantly and independently higher monocyte percentages than carriers of the TT genotype. Conversely, carriers of the allele T had significantly higher eosinophil percentages, although this result was not statistically independent.

Moreover, in the group of infants who did not develop ROP, carriers of allele C of the ***TH*** polymorphism had significantly and independently higher total bilirubin values compared to carriers of the TT genotype.

### 2.5. BDNF Polymorphism, Hematological and Biochemical Parameters, and Incidence of ROP

For mean corpuscular volume (MCV) and MCH, significantly and independently lower values were observed in carriers of the genotype GG of the *BDNF* polymorphism, but only in infants who did not develop ROP (Table 5). In this same group, carriers of the genotype GG also had statistically lower median neutrophil and eosinophil percentages, along with higher median lymphocyte percentages. Similar patterns were found in the group of infants who developed ROP for carriers of the allele G.

In infants who did not develop ROP, the genotype GG of the ***BDNF*** polymorphism was associated with significantly but not independently lower leukocyte counts. Conversely, in the group of infants who developed ROP, the genotype GG was associated with significantly but not independently higher monocyte counts. No notable differences in biochemical parameters were observed among the groups.

### 2.6. EPO Polymorphism, Hematological and Biochemical Parameters, and Incidence of ROP

A significantly lower neutrophil percentages was associated with the genotype GG of the ***EPO*** polymorphism, but only in infants who did not develop ROP (Table 6). In contrast, among infants who developed ROP, the allele G of the same polymorphism was associated with significantly higher erythrocyte counts and lower MCH, reticulocyte, and MCV values, though the latter two parameters were not independent. Additionally, in this group, the genotype GG of ***EPO*** was associated with reduced plateletcrit and increased LMR. Regarding biochemical parameters, no significant differences were found.

### 2.7. Epistatic Relationships and ROP

Positive epistasis was found between the genotype GG of the ***NGF*** polymorphism and the allele G of the *BDNF* polymorphism, suggesting that this combination is associated with a 1.79-fold risk of developing ROP (Table 7). Additionally, positive epistasis was observed between the genotype GG of the *NGF* polymorphism, the allele G of the *BDNF* polymorphism, and the allele C of the *TH* polymorphism, suggesting a 2.38-fold increased risk of developing ROP.

### 2.8. Epistatic Relationships, Hematological Parameters, and ROP

The epistatic relationships between genetic polymorphisms and hematological parameters in relation to ROP outcomes are presented in Table 8, Table 9 and Table 10. In preterm infants who did not develop ROP, carriers of both the *NGF* (GG) and *BDNF* (allele G) polymorphisms, as well as carriers of the *NGF* (GG), *BDNF* (allele G), and *TH* (allele C) polymorphisms, had significantly and independently lower percentages of immature granulocytes compared to carriers of other genotypes of the same genes (Table 8 and Table 10). However, in the group that developed ROP, this relationship was reversed. Carriers of these genotypes exhibited higher values of immature granulocytes than those with other genotypes, although the difference was not statistically significant.

A significantly and independently greater percentage of lymphocytes and monocytes was observed in carriers of both the *BDNF* (allele G) and *TH* (allele C) polymorphisms compared to carriers of other genotypes of these genes, but only in the group that developed ROP (Table 8). Conversely, in simultaneous carriers of the *NGF* (GG) and *TH* (allele C) polymorphisms, lymphocyte and monocyte counts were significantly and independently lower than those in carriers of other genotypes, also within the group that developed ROP (Table 8).

Among infants who developed ROP, those carrying both the *EPO* (GG) and *BDNF* (allele G) polymorphisms exhibited significantly and independently lower platelet counts and plateletcrit (Table 9).

Conversely, in infants without ROP, carriers of the *BDNF* (allele G) and *TH* (allele C) polymorphisms showed a significant reduction in lymphocyte count and the percentage of immature granulocytes. However, this association lost significance after adjusting for GA and the number of RBC transfusions, although it remained a trend for immature granulocytes.

In the ROP-affected group, carriers of the *EPO* (GG) and *TH* (allele C) polymorphisms showed a reduction in monocyte and platelet counts, as well as plateletcrit, and an increase in the LMR. These results were statistically significant but not independent.

For carriers of the *NGF* (GG), *EPO* (GG), and *BDNF* (allele G) polymorphisms, there was a significant reduction in platelecrit exclusively in the group that developed ROP (Table 10), though these results were also not statistically independent.

A significantly lower total eosinophil count was observed for simultaneous carriers of the *EPO* (GG), *BDNF* (allele G), and *TH* (allele C) polymorphisms compared to carriers of other genotypes of these genes in both groups of infants (Table 10). This result was statistically independent only in the ROP group.

Finally, for patients who were simultaneous carriers of *EPO* (GG), *BDNF* (allele G), and *TH* (allele C), the plateletcrit was significantly but not independently lower only in the group that developed ROP.

## 3. Discussion

This study reaffirms that the incidence of ROP is higher in preterm infants with lower GA and BW, consistent with previous findings and existing literature [31,32,33]. Our results emphasize that a lower GA is associated with a greater risk of ROP and other prematurity-related complications, aligning with other studies in the field [34]. Multivariate regression analysis revealed that GA and the number of RBC transfusions significantly contributed to the development of ROP, highlighting their critical role in its pathogenesis, consistent with both the existing literature and our past research [35].

Significant differences in hematological parameters were observed between infants who developed ROP and those who did not, aligning with our previous findings [31,32]. Elevated erythroblasts, RDW, basophils, and lower platelet counts were independently associated with ROP. These hematological abnormalities may reflect physiological challenges in preterm infants, such as impaired kidney function and reduced EPO production, contributing to anemia in early life [36]. Elevated erythroblasts and RDW suggest compensatory erythropoietic activity in response to anemia. Reduced platelet counts could be linked to lower EPO levels, which not only drive erythropoiesis but also support thrombopoiesis and offer cytoprotective effects on platelets [37].

Further, we explored the influence of functional genetic polymorphisms in *NGF* (rs6330), *BDNF* (rs7934165), *TH* (rs10770141), and *EPO* (rs507392) on the modulation of hematological and biochemical parameters during the first week of life, as well as their potential association with ROP development. Our aim was to elucidate how these genes, which are involved in retinal neurodevelopment, vascularization, homeostasis, energy balance, and stress response, modulate the intermediate phenotype (hematological and biochemical biomarkers) and the distant phenotype (ROP).

For clarity and practicality, we will present the discussion of the results for each genetic polymorphism studied sequentially, examining their relationships with intermediate and distant phenotypes, followed by an analysis of epistatic relationships. The discussion concludes with a brief consideration of the study’s limitations.

### 3.1. Nerve Growth Factor

The *NGF* rs6330 polymorphism (genotype GG) was significantly and independently associated with increased susceptibility to ROP in our cohort. *NGF* is a crucial neurotrophin involved in the survival and growth of neurons, including peripheral sensory, sympathetic, and cholinergic neurons [38]. *NGF* supports neuropeptide and neurotransmitter synthesis, neuron morphology, and axonal growth [39] and is expressed throughout the visual pathway, spanning the retina to the visual cortex [24]. In the retina, *NGF* is produced by retinal pigment epithelium, ganglion cells, and Müller cells, and interacts with these cells as well as with extraneuronal targets such as endothelial and immune cells [38,40].

The *NGF* gene, located on chromosome 1p13.1, spans approximately 52.3 kb and comprises three exons [41]. NGF is initially produced as a 26–32 kDa precursor (pro-NGF) that undergoes processing into a 13 kDa mature form [42]. Its biological effects are mediated by two types of receptors: the high-affinity tropomyosin receptor kinase A (TrkA) receptor and the low-affinity p75 neurotrophin receptor (p75NTR) [43]. TrkA primarily mediates NGF’s neurotrophic effects via several pathways, including the ras-mitogen-activated protein kinase (RAS-MAPK) pathway, extracellular signal-regulated kinase (ERK), phosphatidylinositol 3-kinase-Akt (PI3K), and phospholipase C-gamma (PLC-γ) [42]. Furthermore, NGF upregulates the expression of the Krüppel-like factor 4 (KLF4) transcription factor, which is crucial for initiating anti-inflammatory and anti-apoptotic responses [44]. The ERK5/KLF4 signaling cascade has been suggested as a key mechanism by which NGF induces neuroprotection against oxidative stress [44].

Following binding, the NGF/TrkA complex is retrogradely transported to the neuronal cell body, influencing transcriptional responses distinct from those triggered by cell surface signaling [40]. Additionally, NGF binding to p75NTR modulates TrkA signaling and activates alternative pathways, including Jun N-terminal kinases (JNKs), nuclear factor kappa B (NF-κB), and ceramide production. Without co-expressed TrkA, these pathways can lead to apoptosis [42].

Our study reveals, for the first time, a significant association between the genotype GG of the *NGF* polymorphism (rs6330) and susceptibility to ROP, with the minor allele A potentially offering a protective effect. Previous studies have shown that the minor allele (A) of rs6330 is associated with reduced plasma levels of *NGF* [45], potentially influencing its neurotrophic, anti-apoptotic, angiogenic, and immunoregulatory functions. This polymorphism, involving an alanine-to-valine substitution at position 35 of the pro-NGF, may influence NGF processing and secretion [41,46]. It appears to impact the balance between NGF and pro-NGF levels, affecting neuronal homeostasis and post-translational processing [46].

The genotype GG of the *NGF* polymorphism has been linked to various neurovascular and psychiatric conditions [47]. In the context of ROP, NGF’s anti-apoptotic effects may increase susceptibility. Previous research indicates that NGF can exert a pro-angiogenic effect by reducing endothelial cell apoptosis in hypoxic conditions [48].

Our study further substantiates the association between the genotype GG of the *NGF* polymorphism and the development of ROP, suggesting that the minor allele A may offer a protective effect. This polymorphism seems to influence NGF protein processing and secretion, thereby affecting neurodevelopment and vascular functions [41]. Carriers of the genotype GG exhibited a higher incidence of ROP, likely due to NGF’s anti-apoptotic effects in hypoxic environments, which may exacerbate retinal neovascularization [48].

Regarding biochemical parameters associated with the *NGF* polymorphism (Table 3), CRP levels were significantly and independently lower in carriers of the allele G who did not develop ROP. Conversely, CRP levels were elevated in allele G carriers with ROP. Notably, in the ROP group, allele G carriers of the NGF polymorphism exhibited significantly lower total bilirubin levels compared to AA carriers, in contrast to the inverse relationship observed in the non-ROP group. This pattern suggests that allele G carriers with ROP may experience a more pronounced inflammatory response than their AA counterparts, highlighting the modulation of inflammation by NGF due to its regulatory role in immune cell activity [38,43].

In the ROP cohort, the increased inflammation associated with the allele G, which may impair RBC production, could explain the reduced bilirubin levels observed, as bilirubin is a byproduct of RBC hemolysis. Additionally, given the antioxidant properties of bilirubin [49], its reduction might exacerbate the risk of ROP.

The interplay between NGF’s immunoregulatory functions and angiogenic potential underscores its dual role in ROP development. While NGF may support vascular repair and neuroprotection, its pro-angiogenic and inflammatory modulation under hypoxic stress could drive pathological neovascularization. Further research is needed to elucidate how NGF-driven immune responses, particularly T cell activity, contribute to ROP progression.

### 3.2. Tyrosine Hydroxylase

NGF regulates norepinephrine production in sympathetic neurons by inducing the transcription of TH, the rate-limiting enzyme in catecholamine synthesis [43]. ***TH***, located on chromosome 11p15.5 and consisting of 13 exons, is essential for converting L-tyrosine into dihydroxyphenylalanine (L-DOPA), the precursor of catecholamines such as dopamine, norepinephrine, and epinephrine [50].

Dopamine serves as a crucial neurotransmitter within the brain, contributing to numerous central nervous system functions. It is also the predominant catecholamine in the retina across vertebrate species, including humans [51].

TH activity influences sympathetic nervous system (SNS) function, regulating catecholamine secretion and systemic effects [13]. Through TH-mediated catecholamine synthesis, the SNS regulates responses such as cold-induced stress and energy balance by activating UCP1 in BAT, which promotes thermogenesis [52,53]. BAT is characterized by its high mitochondrial content and dense sympathetic innervation, making it a metabolically active organ crucial for maintaining body temperature [54]. This energy-intensive process requires substantial oxygen consumption, potentially linking BAT activity to systemic oxygen homeostasis.

Reduced BAT activity may exacerbate hyperoxia, a key factor in the first phase of ROP, while increased thermogenic demand could contribute to hypoxia, potentially driving mechanisms underlying the second phase. These observations suggest that genetic regulation of BAT activity, possibly influenced by the *TH* (rs10770141) polymorphism, could indirectly affect retinal vascular dynamics and influence ROP development. Further research is needed to elucidate these interactions and their physiological significance.

The *TH* polymorphism rs10770141 (C-824T), located in the promoter region, affects transcription factor binding and is associated with increased norepinephrine excretion and blood pressure in response to cold stress [13]. *TH* expression and activity are regulated at multiple levels—transcriptional, translational, and post-translational. Additionally, TH is subject to feedback inhibition by catecholamines and regulation through phosphorylation mechanisms. Dopamine not only inhibits TH feedback but also stabilizes TH, particularly in the axon-terminal compartment, aiding in the maintenance of TH levels [55].

Genetic variations in *TH* influence SNS activity and have been linked to conditions such as hypertension, bipolar disorder, and Parkinson’s disease [13,56,57].

In preterm infants who did not develop ROP, the genotype CC of the *TH* polymorphism (rs10770141) was associated with significantly lower MCH, while carriers of the allele C exhibited higher total bilirubin levels compared to those with the allele T (Table 4). The allele T is related to increased catecholamine secretion [14]. Our findings suggest that individuals carrying the allele C likely exhibit reduced catecholamine levels and potentially lower EPO production. Reduced EPO results in decreased erythropoiesis and hemoglobin synthesis, leading to lower MCH, while diminished anti-apoptotic effects cause increased RBC destruction and elevated total bilirubin.

In the ROP cohort, carriers of the allele C exhibited significantly higher monocyte percentages compared to genotype TT individuals, while eosinophil percentages were elevated in carriers of the allele T compared to those with the genotype CC. These findings highlight the influence of the *TH* polymorphism on sympathetic regulation of immune cell recruitment and function [53,58].

Additionally, in the ROP group, homozygotes (CC + TT) for the ***TH*** polymorphism exhibited higher neutrophil percentages and NLR than heterozygotes. Homozygous CC individuals, who produce fewer catecholamines, demonstrated reduced lymphocyte and eosinophil percentages, likely due to diminished sympathetic signaling, which normally facilitates recruitment of these cells. Conversely, the genotype TT, associated with increased sympatho-adrenergic activation and chronic stress, was linked to elevated neutrophil percentages and NLR, while potentially impairing eosinophil recruitment and sympathetic innervation [58,59].

Our findings reveal that the *TH* polymorphism rs10770141 (C-824T) is associated with altered NLR, particularly in preterm infants diagnosed with ROP. Elevated NLR has been implicated in increased inflammatory activity, with neutrophils potentially contributing to retinal tissue damage through pro-inflammatory mechanisms, as suggested by their involvement in systemic inflammatory responses [60]. In the context of ROP, these processes may exacerbate retinal neurovascular instability during the disease’s early phase. The association of the genotype CC of the *TH* polymorphism with increased neutrophils and NLR suggests that reduced catecholamine-mediated anti-inflammatory effects may intensify neutrophil-driven damage. Conversely, reduced lymphocytes in homozygotes (CC) might reflect impaired regulatory T cell (Treg) activity, critical in modulating inflammatory responses and maintaining tissue homeostasis. These findings highlight the influence of genetic regulation of catecholamine synthesis on inflammation and its impact on retinal vascularization.

Building on these findings, our data suggest that the *TH* polymorphism modulates not only SNS activity but also hematological parameters and immune responses, particularly in relation to ROP development in preterm infants. These insights underscore the potential clinical relevance of TH in influencing the balance between inflammatory and neurovascular mechanisms in ROP.

### 3.3. Brain-Derived Neurotrophic Factor

BDNF promotes neuronal growth, synaptogenesis, and neuroplasticity [10]. Located on chromosome 11p14 and containing 12 exons with specific promoters, the *BDNF* gene is highly methylated and regulated epigenetically [10,61]. The minor allele (A) of the *BDNF* rs7934165 polymorphism is associated with increased methylation at several CpG sites, potentially leading to reduced gene expression [61]. This epigenetic modification may decrease BDNF protein availability, which could impact its neurotrophic and immunomodulatory functions.

BDNF is initially produced as pre-pro-BDNF and cleaved into its mature form, which interacts with the TrkB receptor to support synaptic plasticity, while pro-BDNF binds to p75NTR, leading to apoptosis and axonal retraction [10,62]. BDNF also has anti-inflammatory effects mediated through ERK and PI3K/AKT pathways and supports lymphocyte proliferation and survival [63].

The reduction in circulating BDNF levels has been associated with the development of severe ROP in multiple studies [4,64,65]. Additionally, a comprehensive study on candidate genes identified variants within the *BDNF* intron (rs7934165 and rs2049046) as being associated with severe ROP [66].

In our study, the *BDNF* rs7934165 polymorphism was significantly associated with neutrophil and lymphocyte percentages. Carriers of the allele G exhibited higher lymphocyte percentages and lower neutrophil percentages in ROP cases, likely reflecting the higher BDNF expression linked to this allele, which may enhance its anti-inflammatory effects. Conversely, the association between allele A and higher neutrophil percentages in non-ROP cases may reflect a diminished capacity of BDNF to regulate inflammatory responses and immune cell activity, particularly neutrophils, under baseline conditions, potentially due to reduced BDNF expression linked to this allele.

These findings underscore the dual role of BDNF in modulating inflammatory pathways, where higher expression linked to the allele G may promote neuroprotection and immune regulation, while reduced expression associated with the allele A may impair these processes. This balance could influence the progression of ROP, particularly in the context of persistent inflammation or oxidative stress.

### 3.4. Erythropoietin

The *EPO* gene, located on chromosome 7q21, encodes the precursor of EPO, a key angiogenic factor involved in erythropoiesis and retinal angiogenesis [67,68]. EPO plays a crucial role in developing retinal vasculature, with low serum levels potentially impairing early angiogenesis in ROP, while elevated levels may promote pathological neovascularization during later stages [68].

In our cohort, the *EPO* (rs507392) polymorphism was significantly associated with reduced MCH, reticulocyte count, and MCV in preterm infants diagnosed with ROP. These changes are likely attributed to reduced EPO expression and impaired erythropoiesis. Moreover, this polymorphism was correlated with lower plateletcrit and increased LMR, also likely resulting from reduced EPO levels.

Although no direct association was identified between the *EPO* polymorphism and ROP (distant phenotype), the polymorphism influenced some hematological parameters related to anemia and thrombocytopenia (intermediate phenotypes), both recognized as potential risk factors for ROP [69,70]. Previous studies have reported variable findings regarding the relationship between this *EPO* polymorphism and diabetic retinopathy [67,71].

### 3.5. Epistatic Relationships

The genetic underpinnings of complex diseases often arise from the interplay of multiple genes (epistasis) and environmental factors. We identified a significant positive epistatic interaction between the *NGF* (genotype GG) and *BDNF* (allele G) polymorphisms in relation to ROP risk (Table 7). This interaction was more pronounced and remained independent when the *TH* (allele C) was also considered. Despite the neuroprotective role of NGF, it exhibits pro-angiogenic and anti-apoptotic effects [48]. Similarly, BDNF exhibits neuroprotective and anti-apoptotic functions, though it also demonstrates angiogenic properties [72], which are likely more pronounced in carriers of the allele G. We hypothesize that the combined anti-apoptotic actions of both neurotrophins may increase susceptibility to ROP.

Carriers of the *NGF* (GG), *BDNF* (allele G), and *TH* (allele C) genotypes exhibited a significantly higher risk of developing ROP. NGF regulates catecholamine production via TH [43], and the *TH* (allele C) is linked to reduced catecholamine secretion, including dopamine, which is essential for retinal function [51]. This catecholamine reduction could disrupt the balance between angiogenic and anti-angiogenic factors, potentially influencing ROP development. While dopamine has antiangiogenic properties [73], the combined effects of these genotypes might tilt the balance in favor of angiogenesis. Additionally, the reduced formation of BAT linked to *TH* (allele C) may contribute to hyperoxia, further increasing the risk of ROP.

Epistatic interactions between carriers of *NGF* (GG) and *BDNF* (allele G), as well as between carriers of *NGF* (GG), *BDNF* (allele G), and *TH* (allele C), demonstrated significant differences in the percentages of immature granulocytes between infants who developed ROP and those who did not (Table 8 and Table 10). These differences likely reflect impaired inflammatory control in these genotype combinations [43,74,75]. Moreover, the reduced BAT formation associated with *TH* (allele C) may lead to diminished oxygen consumption, a key driver for stimulating EPO production. Since EPO has anti-inflammatory properties, its decrease could further explain the associations observed with *TH* (allele C). This hypothesis warrants further investigation, particularly with the measurement of EPO levels.

Within the ROP group, carriers of both *BDNF* (allele G) and *TH* (allele C) exhibited higher lymphocyte and monocyte percentages (Table 8), possibly due to BDNF’s enhanced anti-inflammatory effects. Conversely, *NGF* (GG) and *TH* (allele C) carriers had lower lymphocyte and monocyte counts, potentially resulting from reduced catecholamine production, leading to diminished sympathetic stimulation of adipose tissue.

Preterm infants with the *EPO* (GG) genotype had significantly lower plateletcrit, but only within the ROP group (Table 6), likely due to reduced EPO secretion, which has protective, anti-apoptotic effects on platelets. Epistatic analysis revealed positive interactions between *EPO* (GG) and *BDNF* (allele G) (Table 9), with significant reductions in platelet counts and plateletcrit in ROP infants carrying these genotypes. While this combination does not directly correlate with ROP development, it shows a significant epistatic relationship with intermediate phenotypes like reduced platelet counts and plateletcrit, linked to higher ROP risk.

Carriers of the *EPO* (GG), *BDNF* (allele G), and *TH* (allele C) genotypes showed significantly lower eosinophil levels, an independent association only in the ROP group (Table 10). This may stem from reduced catecholamine synthesis linked to *TH* (allele C), which impairs BAT formation. Sympathetic stimulation is crucial for recruiting lymphocytes and eosinophils involved in BAT development [53].

EPO receptors are abundant in adipocytes and macrophages of white adipose tissue, highlighting its role in metabolic homeostasis [76]. EPO also promotes brown adipose cell differentiation and regulates UCP1 expression, essential for thermogenesis [19,29].

### 3.6. Clinical Implications and Future Directions

#### 3.6.1. Clinical Implications

Our findings highlight the potential role of genetic polymorphisms in *NGF*, *BDNF*, *TH*, and *EPO* in modulating the pathophysiology of ROP through their influence on hematological, neurovascular, and inflammatory pathways. These insights could inform the development of genetic risk models to identify preterm infants at higher risk for ROP, enabling early intervention strategies. For instance, the association of *TH* and *BDNF* polymorphisms with elevated NLR highlights the importance of addressing neutrophil-mediated inflammation, which may contribute to retinal neurovascular instability. Similarly, altered lymphocyte profiles and eosinophil counts associated with these polymorphisms emphasize the need for targeted inflammatory monitoring in preterm infants. The *NGF* polymorphism’s dual role in angiogenesis and inflammation suggests its potential as a biomarker for stratifying risk in the different phases of ROP.

#### 3.6.2. Future Research Directions

Future studies should explore gene–environment interactions, such as how neonatal care practices, oxygen therapy, blood transfusions, or maternal stress influence the expression and effects of these polymorphisms. Expanding cohort sizes across diverse populations will be essential to validate these associations and ensure generalizability. Mechanistic studies focusing on NGF and BDNF pathways may elucidate therapeutic targets for modulating inflammation or promoting neurovascular repair. Furthermore, longitudinal studies could examine the long-term neurodevelopmental outcomes in infants with these genetic polymorphisms, providing a more comprehensive understanding of their clinical impact.

To provide an integrated overview of the genetic, hematological, and epistatic findings, and their relevance to ROP development, a synthesized flowchart is presented (Figure 1). This visual summary highlights the interplay of key pathways, clinical implications, and future research directions discussed in this study, offering a comprehensive perspective on the multifactorial mechanisms underlying ROP.

### 3.7. Limitations of the Study

Finally, it is important to acknowledge the limitations of this study. The lack of serum measurements for NGF, BDNF, and EPO precluded a more direct quantitative analysis of these molecules in relation to the functional polymorphisms investigated. Additionally, the observational design did not include predefined time points for assessing early hematological and biochemical parameters, which could have provided deeper insights. Lastly, the relatively small sample size limits the generalizability of our findings, underscoring the need for larger-scale studies to validate these associations and identify potential biomarkers for ROP.

## 4. Material and Methods

### 4.1. Population

This observational, prospective, multicenter study was conducted between 19 November, 2018, and 21 July, 2021, across eight neonatal intensive care units in Portugal: Centro Hospitalar Universitário de São João (Porto), Centro Materno Infantil do Norte (Centro Hospitalar Universitário do Porto, Porto), Hospital de Braga (Braga), Hospital da Senhora da Oliveira (Guimarães), Centro Hospitalar Universitário de Lisboa Norte (Lisboa), Hospital Prof. Doutor Fernando Fonseca (Amadora), Maternidade Daniel de Matos, and Maternidade Bissaya Barreto (Centro Hospitalar Universitário de Coimbra, Coimbra). This study is registered under ISRCTN16889608.

Inclusion criteria comprised preterm infants who (1) were born before 32 weeks of GA or (2) had a BW of less than 1500 grams, regardless of race or sex. Exclusion criteria included preterm infants with (1) major congenital malformations; (2) ophthalmological conditions (acquired or congenital) unrelated to ROP occurring within the first 12 weeks of life, except for congenital nasolacrimal duct obstruction, keratitis, and conjunctivitis; (3) death prior to the first ophthalmological evaluation for ROP screening; (4) insufficient clinical data due to transfer to another hospital; or (5) lack of informed consent from parents or legal guardians.

Based on the specified inclusion and exclusion criteria, the study included a total of 396 preterm infants. Of these, 198 (50.0%) were female, and 158 (39.9%) were diagnosed with ROP.

### 4.2. ROP Screening and Ophthalmological Data Collection

ROP screening was conducted for all preterm infants either at 31 to 33 weeks of postmenstrual age or at 4 to 6 weeks of postnatal age, following the guidelines established by the Portuguese Society of Neonatology [77]. After pharmacological mydriasis was administered, qualified ophthalmologists performed examinations using indirect binocular ophthalmoscopy and/or digital fundus imaging (RetCam). Findings for each eye were meticulously recorded according to the Revisited International Classification of ROP [78]. Subsequent examinations were scheduled based on the initial retinal assessment and continued until full retinal vascularization or ROP remission was observed.

### 4.3. Demographic, Clinical, and Laboratory Data

Data for demographic, clinical, and laboratory analyses were systematically retrieved from patient medical records.

#### 4.3.1. Clinical Data

The clinical data collected included maternal and neonatal information. For maternal data, this encompassed: (1) gestational factors such as the use of assisted reproductive technologies, multiple pregnancies, or twin gestation; (2) prenatal interventions, including the administration of corticosteroids or magnesium sulfate, and the presence of maternal chorioamnionitis. Neonatal data included: (1) birth characteristics such as BW (in grams), GA (in weeks), sex, delivery method, and 5-min Apgar scores; (2) the duration of oxygen therapy (in days); (3) the duration of hyperglycemia (blood glucose levels exceeding 125 mg/dL, in days); (4) the number of transfusions for RBCs and platelets; and (5) associated conditions, including bronchopulmonary dysplasia, necrotizing enterocolitis, periventricular–intraventricular hemorrhage (PIVH), hemodynamically significant patent ductus arteriosus, and periventricular leukomalacia.

#### 4.3.2. Laboratory Parameters

Hematological and biochemical parameters from the first week of life were analyzed using standardized methods at the pathology services of participating hospital centers. Hemograms were performed using automated hematology analyzers, such as Sysmex^®^ (Kobe, Japan) or Beckman Coulter^®^ (Brea, CA, USA), which employ impedance technology and flow cytometry for cell counts and indices. CRP was measured in plasma or serum samples using nephelometric and immunoturbidimetric assays on automated biochemistry analyzers (e.g., Roche Cobas^®^, Basel, Switzerland; Abbott Architect^®^, Abbott Park, IL, USA; Olympus^®^, Tokyo, Japan; Beckman-Coulter^®^, Brea, CA, USA; and Siemens^®^_,_ Erlangen, Germany). Bilirubin levels were determined in plasma or serum samples using the Jendrassik–Grof method on automated analyzers.

To ensure consistency and comparability, all laboratories adhered to strict internal and external quality control programs and harmonized practices aligned with international standards.

The parameters assessed included erythrocytes (×10^12^/L), hemoglobin (g/dL), RDW (%), erythroblast count (×10^9^/L), RDW (%), reticulocytes (%), MCH (pg), MCV (fl), leukocytes (×10^9^/L), neutrophils (×10^9^/L and %), lymphocytes (×10^9^/L and %), NLR, monocytes (×10^9^/L and %), LMR, basophils (%), eosinophils (×10^3^/L and %), immature granulocytes (%), granulocytes (%), platelets (×10^9^/L), plateletcrit (%), platelet distribution width (PDW), direct bilirubin (mg/dL), total bilirubin (mg/dL), and CRP (mg/dL).

### 4.4. Genetic Polymorphism Identification

Genetic analysis was conducted as previously described [32], using DNA extracted from buccal swabs or, when necessary, from surplus blood samples obtained during routine laboratory testing in the first four weeks of life. Genotyping was performed using MicroChip DNA on the high-throughput platform with iPlex MassARRAY^®^ system (Agena Bioscience, San Diego, CA, USA). Specific PCR reactions were designed for each allelic variant, and genotypes were determined using MALDI-TOF mass spectrometry. The resulting genotyping data were analyzed with HeartGenetics’ EARTDECODE^®^ software. Genotypic and allele frequencies were calculated, and their distribution was tested for compliance with Hardy–Weinberg equilibrium in the group of preterm infants who did not develop ROP (No ROP group).

### 4.5. Statistical Analysis

The normality of the variables was assessed using the Kolmogorov‒Smirnov test, with values presented as medians and interquartile ranges. The Pearson χ^2^ test was applied to assess the significant differences between groups. The Mann‒Whitney test was used to compare groups. Logistic regression analysis and the corresponding 95% CIs were determined with a binary dependent variable to model the probability of a risk factor for no ROP/ROP and genetic polymorphisms. A multivariate regression analysis (backward conditional) was performed on variables identified as significant in univariate analysis and those deemed clinically relevant. In the multivariate analysis, GA and the number of RBC transfusions were identified as significant risk factors and all results were adjusted accordingly. The statistical analysis was performed using the SPSS program version 28.0, with an alpha level for all tests of 0.05.

### 4.6. Ethics Approval

This research adhered to the ethical principles established in the Declaration of Helsinki and its amendments. The study protocol, data collection tools, and informed consent procedures received approval from the Scientific Council at the Faculty of Medicine, University of Lisbon, as well as the Ethics Committees of all collaborating hospital centers. Specific details regarding the institutional approvals are included in the Institutional Review Board Statement. Written informed consent was obtained from the parents or legal guardians of all participants, and all data were anonymized and managed with strict confidentially.

## 5. Conclusions

In conclusion, our study suggests for the first time that the genotype GG of the NGF (rs6330) polymorphism may increase the risk of developing ROP. Additionally, our findings suggest that polymorphisms in genes associated with neurodevelopment, angiogenesis, homeostasis, neuroendocrine, and neuro-immune pathways might contribute to the etiopathogenesis of ROP. The hematological findings associated with these polymorphisms highlight potential mechanisms by which genetic variations modulate inflammatory responses in ROP. Elevated NLR, observed in association with TH and BDNF polymorphisms, underscores the role of neutrophil-mediated tissue damage in retinal neurovascular instability. This is further compounded by altered lymphocyte profiles, which may reflect impaired regulatory immune functions. The interplay between pro-inflammatory neutrophils, altered lymphocyte profiles, and eosinophils warrants further investigation, particularly in the context of genetic predisposition to inflammation-driven retinal damage. These insights could inform strategies for early identification and targeted modulation of inflammatory pathways in preterm infants at risk of ROP.

Replication studies in diverse populations are necessary to validate our results. Investigating the correlation of these genes with epigenetic factors could also enhance our understanding of ROP. Future research should include a broader range of genetic polymorphisms to capture more variability in genes related to the neurotrophin system.

The associations between the studied polymorphisms and ROP (distant phenotype), as well as related biomarkers (intermediate phenotype), could provide insights into the etiological mechanisms of ROP and help identify new biomarkers. If confirmed, these findings may significantly advance our understanding of ROP’s etiopathogenesis and contribute to the development of improved screening and therapeutic strategies.

## Figures and Tables

**Figure 1 ijms-26-00898-f001:**
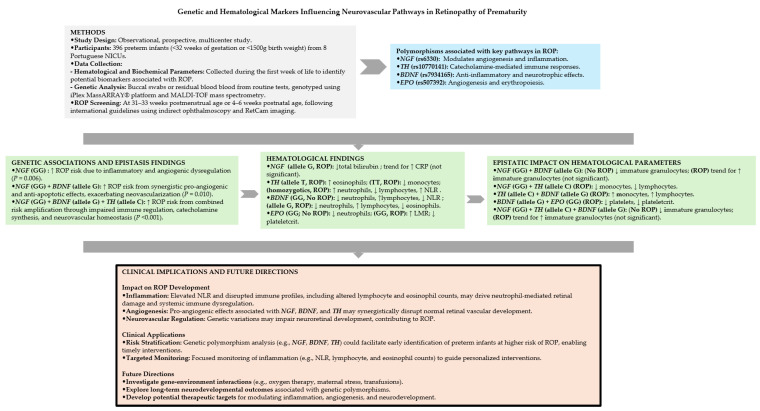
Schematic overview of study design, genetic associations, and ROP implications. *BDNF*, *brain-derived neurotrophic factor*; *EPO*, *erythropoietin*; *NGF*, *nerve growth factor*; LMR, lymphocyte-monocyte ratio; NLR, neutrophil/lymphocyte ratio; ROP, retinopathy of prematurity; *TH*, *tyrosine hydroxylase*.

**Table 1 ijms-26-00898-t001:** Clinical characteristics of preterm infants according to the development of ROP (no ROP or ROP).

Clinical Characteristics	No ROP *n* (%) or Median, Interquartile Range	ROP *n* (%) or Median, Interquartile Range	*p*	*p**
Number of individuals, *n* (%)	238	158	n.a.	n.a.
Gestational age (weeks)	30.4, 2.3	28.0, 2.9	**<0.001**	n.a.
Birth weight (g)	1300.0, 381.0	980.0, 434.3	**<0.001**	**0.015**
Gender: Male Female	126 (52.9) 112 (47.1)	72 (45.6) 86 (54.4)	0.182	0.157
Apgar minute 5: < 7	13 (5.5)	24 (15.6)	**0.001**	0.220
Assisted reproductive technologies	26 (10.9)	22(13.9)	0.753	0.757
Maternal chorioamnionitis	23 (9.7)	20(12.7)	0.328	0.989
Twin or multiple births	77 (32.4)	35 (22.2)	**0.040**	0.220
Type of delivery (Cesarean)	165 (69.3)	104 (65.8)	0.538	0.691
Necrotizing enterocolitis	9 (3.8)	14 (8.9)	**0.045**	0.820
Bronchopulmonary dysplasia	22 (9.2)	59 (37.3)	**<0.001**	0.923
Patent ductus arteriosus	15 (6.3)	40 (25,3)	**<0.001**	0.203
Periventricular/intraventricular hemorrhage	16 (6.7)	31 (19.6)	**<0.001**	0.487
Cystic periventricular leukomalacia	4 (1.7)	8 (5.1)	0.071	0.655
Sepsis	77 (32.4)	104 (65.8)	**<0.001**	0.167
Glycemia ≥ 125 mg/dL (≥4 days)	34 (14.3)	60 (38.0)	**<0.001**	0.731
Prenatal steroids	223 (93.7)	143 (90.5)	0.332	0.172
Prenatal magnesium	81 (34.0)	66 (41.8)	0.168	0.281
Invasive mechanical ventilation (days)	0.0, 1	3.0, 18	**<0.001**	0.094
Noninvasive mechanical ventilation (days)	3.0, 9	20.0, 29	**<0.001**	0.300
Administration of EPO or darbepoetin	8 (3.4)	11 (7.0)	0.147	0.772
RBC transfusions	43 (18.1)	102 (64.6)	**<0.001**	0.096
Number of RBC transfusions	0.0, 0	1.0, 4	**<0.001**	n.a.
Platelet transfusions	6 (2.5)	35 (22.2)	**<0.001**	**0.004**
Days of platelet transfusions	0.01, 0	0.04, 0	**<0.001**	0.056
Erythrocytes (×10^12^/L)	4.3, 0.9	4.0, 0.8	**<0.001**	0.105
Hemoglobin (g/dL)	16.1, 3.1	14.8, 3.4	**<0.001**	0.177
MCV (fl)	107.4, 9.6	108.3, 12.1	**0.033**	**0.034**
MCH (pg)	37.7, 3.3	37.3, 4.0	0.146	0.559
RDW (%)	16.5, 2.2	17.1, 2.9	**0.001**	**0.033**
Erytroblasts (×10^9^/L)	7.8, 15.8	14.8, 32.0	**<0.001**	**0.015**
Reticulocytes (%)	6.7, 4.1	7.0, 4.9	0.773	0.854
Leukocytes (×10^9^/L)	9.2, 4.8	9.4, 6.8	0.575	0.956
Neutrophils (×10^9^/L)	3.9, 3.3	4.6, 5.4	0.359	0.724
Lymphocytes (×10^9^/L)	4.0, 1.9	3.5, 1.6	**0.037**	0.585
NLR	1.0, 0.9	1.1, 1.3	**0.025**	0.872
Basophils (%)	0.5, 0.4	0.7, 0.6	**0.006**	**0.017**
Eosinophils (%)	2.1, 2.5	1.9, 2.0	0.086	0.108
Monocytes (×10^9^/L)	1.1, 0.8	1.2, 1.1	0.943	0.661
LMR	3.3, 2.6	3.1, 2.6	0.149	0.981
Immature granulocytes (%)	1.2, 1.1	0.9, 0.7	**0.048**	0.073
Platelets (×10^9^/L)	222.2, 110.2	202.0, 121.1	**0.004**	**0.022**
Plateletcrit (%)	0.3, 0.1	0.2, 0.1	**0.002**	0.936
PDW (%)	10.3, 1.6	9.6, 2.1	**0.013**	**0.006**
Direct bilirubin (mg/dL)	0.5, 0.3	0.7, 0.4	**0.005**	0.335
CRP	0.5, 2.5	0.7, 2.2	**0.304**	0.574

CRP, C-reactive protein; LMR, lymphocyte/monocyte ratio; MCH, mean corpuscular hemoglobin; MCV, mean corpuscular volume; n.a., not applicable; NLR, neutrophil/lymphocyte ratio; PDW, platelet distribution width; *p*, *p*-value; *p**, *p*-value adjusted for gestational age and the number of red blood cell transfusions; RDW, red blood cell distribution width; ROP, retinopathy of prematurity. *p*-values less than 0.05 are in bold. Mann‒Whitney test or Pearson χ^2^ test.

**Table 2 ijms-26-00898-t002:** Distribution of genetic polymorphisms (***NGF***, ***TH***, ***BDNF***, and ***EPO***) in the studied population according to the development of ROP (no ROP vs. ROP).

Polymorphism	Genotype	No ROP *N* (%)	ROP *N* (%)	OR [CI 95%]	*p*	*p**
*NGF* (rs6330)	AA	49 (20.7)	21 (13.3)	n.a.	0.061	0.107
	Allele G	188 (79.3)	137 (86.7)	n.a.		
	GG	81 (34.2)	76 (48.1)	1.79 [1.18–2.69]	**0.006**	**0.049**
	Allele A	156 (65.8)	82 (51.9)	1		
	GG + AA	130 (54.9)	97 (61.4)	n.a.	0.213	0.490
	GA	107 (45.1)	61 (38.6)	n.a.		
*TH* (rs10770141)	TT	91 (39.9)	55 (35.0)	n.a.	0.339	0.354
	Allele C	137 (60.1)	102 (65.0)	n.a.		
	CC	33 (14.5)	32 (20.4)	n.a.	0.131	0.430
	Allele T	195 (85.5)	125 (79.6)	n.a.		
	TT + CC	124 (54.4)	87 (55.4)	n.a.	0.917	0.741
	CT	104 (45.6)	70 (44.6)	n.a.		
*BDNF* (rs7934165)	AA	52 (22.1)	28 (17.7)	n.a.	0.309	0.671
	Allele G	183 (77.9)	130 (82.3)	n.a.		
	GG	59 (25.1)	39 (24.7)	n.a.	0.924	0.824
	Allele A	176 (74.9)	119 (75.3)	n.a.		
	GG + AA	111 (47.2)	67 (42.4)	n.a.	0.354	0.591
	GA	124 (52.8)	91 (57.67)	n.a.		
*EPO* (rs507392)	AA	98 (41.9)	54 (34.4)	n.a.	0.140	0.201
	Allele G	136 (58.1)	103 (65.6)	n.a.		
	GG	27 (11.5)	22 (14.0)	n.a.	0.534	0.515
	Allele A	207 (88.5)	135 (86.0)	n.a.		
	GG + AA	125 (53.4)	76 (48.4)	n.a.	0.354	0.427
	GA	109 (46.6)	81 (51.6)	n.a.		

*BDNF*, *brain-derived neurotrophic factor*; *EPO*, *erythropoietin*; n.a., not applicable; *NGF*, *nerve growth factor*; OR, odds ratio; *p*, *p*-value; *p**, *p*-value adjusted for gestational age and the number of red blood cell transfusions. *TH*, *tyrosine hydroxylase*; ROP, retinopathy of prematurity. *p*-values less than 0.05 are in bold. Multivariable logistic regression analysis.

**Table 3 ijms-26-00898-t003:** Relationships of the ***NGF*** polymorphism with hematological and biochemical parameters: (**a**) in the group of infants who did not develop ROP; (**b**) in the group of infants who developed ROP.

**(a) *NGF* (rs6330) Polymorphism in the No ROP Group**												
**Median, Interquartile Range**	**AA**	**Allele G**	** *p* **	** *p** **	**GG**	**Allele A**	** *p* **	** *p** **	**AA + GG**	**GA**	** *p* **	** *p** **
MCV (fl)	106.7, 10.0	107.7, 8.9	0.200	0.128	107.7, 10.5	107.2, 9.2	0.971	0.971	107.1, 10.3	108.2, 8.0	0.306	0.204
LMR	4.1, 2.5	3.3, 2.6	**0.039**	0.084	3.3, 2.7	3.5, 2.4	0.743	0.999	3.5, 2.4	3.1, 2.5	**0.042**	0.875
Total bilirubin (mg/dL)	6.8, 2.1	7.6, 3.4	0.226	0.788	7.6, 3.6	7.1, 2.8	0.315	0.189	7.2, 2.8	7.6, 3.1	0.883	0.377
CRP (mg/dL)	0.7, 9.4	0.4, 1.7	**0.046**	**0.029**	0.3, 2.7	0.5, 1.8	0.766	0.835	0.6, 6.1	0.4, 1.5	0.148	0.072
**(b) *NGF* (rs6330) Polymorphism in the ROP Group**												
**Median, Interquartile Range**	**AA**	**Allele G**	** *p* **	** *p** **	**GG**	**Allele A**	** *p* **	** *p** **	**AA + GG**	**GA**	** *p* **	** *p** **
MCV (fl)	107.5, 7.7	109.0, 13.8	0.209	0.220	108.0, 14.2	109.3, 12.7	0.286	0.328	108.0, 11.2	110.5, 12.9	**0.047**	0.062
LMR	3.8, 3.0	3.0, 2.5	0.622	0.881	3.0, 2.5	3.2, 2.7	0.511	0.877	3.1, 2.7	3.1, 2.7	0.729	0.973
Total bilirubin (mg/dL)	7.4, 2.4	6.4, 2.3	**0.026**	**0.006**	6.3, 2.1	6.6, 2.3	0.147	0.303	6.5, 2.1	6.4, 2.4	0.882	0.292
CRP (mg/dL)	0.5, 1.6	0.7, 2.2	0.722	0.387	0.8, 2.2	0.5, 1.9	0.869	0.926	0.7, 2.2	0.5, 2.1	0.935	0.584

CRP, C-reactive protein; LMR, lymphocyte-monocyte ratio; MCV, mean corpuscular volume; *p*, *p*-value; *p**, *p*-value adjusted for gestational age and the number of red blood cell transfusions. *p*-values less than 0.05 are in bold. ROP, retinopathy of prematurity. Mann‒Whitney test and multivariable logistic regression analysis.

**Table 4 ijms-26-00898-t004:** Relationships of the ***TH*** polymorphism with hematological and biochemical parameters: (a) in the group of infants who did not develop ROP; (b) in the group of infants who developed ROP.

**(a) *TH*** **(rs10770141) Polymorphism in the No ROP Group**												
**Median, Interquartile Range**	**CC**	**Allele T**	** *p* **	** *p** **	**TT**	**Allele C**	** *p* **	** *p** **	**CC + TT**	**CT**	** *p* **	** *p** **
MCH (pg)	36.6, 3.1	38.0, 3.3	**0.008**	**0.006**	38.2, 3.2	37.5, 3.3	0.191	0.322	37.8, 3.1	37.7, 3.5	0.587	0.360
Leucocytes (×10^9^/L)	10.0, 9.7	8.8, 4.4	**0.040**	0.124	9.2, 3.7	9.2, 4.9	0.979	0.327	9.5, 5.3	8.4, 4.6	0.140	0.270
Neutrophiles (%)	44.4, 25.9	40.7, 13.9	0.132	0.232	41.1, 14.9	41.6, 15.0	0.553	0.585	42.0, 16.2	40.7, 13.9	0.628	0.481
Lymphocytes (%)	34.2, 17.9	41.4, 15.7	0.104	0.396	41.3, 19.6	40.3, 16.8	0.270	0.164	40.2, 18.6	42.8, 15.2	0.955	0.467
NLR	1.0, 0.9	1.0, 0.9	0.962	0.808	0.9, 0.8	1.0, 0.9	0.169	0.816	1.0, 0.8	1.0, 0.8	0.187	0.717
Eosinophils (%)	1.8, 2.3	2.2, 2.5	0.268	0.269	2.2, 2.9	2.1, 2.3	0396	0.111	2.1, 2.6	2.2, 2.3	0.965	0.381
Monocytes (%)	4.7, 1.5	4.9, 2.1	0.940	0.729	4.6, 2.4	4.9, 1.2	0.660	0.646	4.6, 2.0	4.9, 1.5	0.606	0.672
Total bilirubin (mg/dL)	7.9, 3.7	7.2, 2.8	0.498	0.335	6.8, 2.4	7.7, 3.2	**0.042**	**0.030**	7.0, 2.9	7.3, 3.0	0.152	0.121
**(b) *TH*** (**rs10770141) Polymorphism in the ROP Group**												
**Median, Interquartile Range**	**CC**	**Allele T**	** *p* **	** *p** **	**TT**	**Allele C**	** *p* **	** *p** **	**CC + TT**	**CT**	** *p* **	** *p** **
MCH (pg)	37.2, 3.9	37.2, 4.2	0.459	0.194	36.9, 5.4	37.4, 3.7	0.493	0.889	36.9, 4.3	37.5, 3.8	0.214	0.232
Leucocytes (×10^9^/L)	11.4, 13.5	8.8, 5.7	0.080	0.310	9.1, 4.8	9.4, 8.2	0.892	0.877	9.7, 7.4	8.6, 6.7	0.128	0.079
Neutrophiles (%)	49.9, 16.5	41.6, 17.7	0.096	0.224	45.6, 19.8	41.6, 17.9	0.195	0.154	47.7, 18.8	40.6, 17.4	**0.010**	**0.021**
Lymphocytes (%)	30.7, 19.6	38.2, 18.6	0.083	0.162	34.7, 23.0	38.8, 19.5	0.172	0.132	33.9, 20.5	41.5, 14.7	**0.007**	**0.012**
NLR	1.7, 1.6	1.1, 1.4	0.081	0.079	1.2, 2.5	1.1, 1.1	0.157	0.369	1.5, 1.9	1.0, 1.0	**0.009**	**0.024**
Eosinophils (%)	1.3, 1.6	2.0, 2.3	**0.038**	0.053	2.0, 2.7	1.8, 1.8	0.153	0.052	1.8, 2.2	1.9, 1.8	0.761	0.735
Monocytes (%)	4.9, 4.0	4.3, 2.1	0.140	0.054	3.6, 1.5	5.0, 2.8	**0.003**	**0.012**	4.2, 2.0	5.0, 2.5	0.162	0.934
Total bilirubin (mg/dL)	6.7, 2.6	6.4, 2.2	0.541	0.130	6.1, 2.1	6.6, 2.2	0.235	0.123	6.3, 2.3	6.6, 2.1	0.512	0.819

MCH, mean corpuscular hemoglobin; NLR, neutrophil/lymphocyte ratio; *p*, *p*-value; *p**, *p*-value adjusted for gestational age and the number of red blood cell transfusions. ROP, retinopathy of prematurity; *TH*, *tyrosine hydroxylase*; *p*-values less than 0.05 are in bold. Mann‒Whitney test and multivariable logistic regression analysis.

**Table 5 ijms-26-00898-t005:** Relationships of the *BDNF* polymorphism with hematological and biochemical parameters: (**a**) in the group of infants who did not develop ROP; (**b**) in the group of infants who developed ROP.

**(a) *BDNF* (rs7934165) Polymorphism in the No ROP Group**												
**Median, Interquartile Range**	**AA**	**Allele G**	** *p* **	** *p** **	**GG**	**Allele A**	** *p* **	** *p** **	**AA + GG**	**GA**	** *p* **	** *p** **
MCV (fl)	106.0, 8.7	107.7, 10.1	0.228	0.249	48; 104.8, 2.8	133; 108.0, 8.5	**0.029**	**0.026**	105.3, 9.4	108.7, 7.3	**0.004**	**0.003**
MCH (pg)	37.4, 2.7	37.8, 3.6	0.715	0.635	34.5, 2.9	38.1, 3.0	**0.007**	**0.017**	37.1, 2.9	38.3, 2.8	**0.007**	**0.012**
Leucocytes (×10^9^)	10.1, 5.1	8.8, 4.3	0.051	0.057	8.0, 3.3	9.7, 4.4	**0.020**	0.239	9.2, 4.9	9.6, 4.5	0.731	0.510
Neutrophils (%)	39.6, 19.8	41.6, 14.1	0.893	0.668	38.4, 13.8	42.8, 16.1	**0.014**	**0.019**	38.8, 15.9	43.3, 13.6	**0.023**	0.110
Lymphocytes (%)	38.4, 19.8	41.0, 15.8	0.208	0.302	44.0, 15.1	39.9, 17.0	**0.033**	0.089	40.8, 18.4	40.4, 15.3	0.438	0.548
NLR	0.9, 1.2	1.0, 0.8	0.418	0.987	0.9, 0.5	1.0, 0.9	**0.022**	**0.012**	0.9, 0.6	1.1, 0.9	**0.008**	**0.041**
Basophils (%)	0.6, 0.4	0.5, 0.4	0.364	0.663	0.5, 0.4	0.5, 0.4	0.133	0.245	0.5, 0.5	0.5, 0.4	0.600	0.515
Eosinophils (%)	2.2, 2.8	2.1, 2.4	0.185	0.249	1.6, 1.7	2.2, 2.8	**0.006**	**0.007**	1.9, 2.2	2.2, 2.8	0.209	0.312
Monocytes (×10^9^)	4.6, 1.3	4.9, 2.1	0.982	0.723	4.9, 0.6	4.8, 2.1	0.825	0.772	4.6, 0.6	4.9, 2.8	0.878	0.787
Immature granulocytes (%)	1.6, 1.4	1.0, 1.0	**0.014**	**0.029**	1.0, 2.0	1.3, 1.0	0.582	0.906	1.5, 1.5	1.0, 1.0	0.089	0.057
PDW (%)	10.1, 0.9	10.4, 1.6	**0.047**	0.092	10.2, 1.8	10.3, 1.4	0.532	0.680	10.1, 1.2	10.5, 1.7	0.280	0.480
Total bilirubin (mg/dL)	8.0, 2.6	6.9, 2.7	**0.009**	0.054	7.4, 4.0	7.2, 2.7	0.571	0.280	7.8, 3.3	6.8, 2.3	**0.006**	**0.013**
**(b) *BDNF* (rs7934165) Polymorphism in the ROP Group**												
**Median, Interquartile Range**	**AA**	**Allele G**	** *p* **	** *p** **	**GG**	**Allele A**	** *p* **	** *p** **	**AA + GG**	**GA**	** *p* **	** *p** **
MCV (fl)	108.2, 11.9	109.0, 13.2	0.825	0.145	110.9, 12.8	108.3, 11.1	0.203	0.260	109.3, 11.4	108.6, 13.0	0.357	0.391
MCH (pg)	37.3, 3.3	37.2, 4.4	0.779	0.641	37.5, 4.5	37.0, 3.9	0.327	0.451	37.4, 3.7	36.9, 4.3	0.537	0.611
Leucocytes (×10^9^)	14.1, 20.3	9.0, 5.6	0.061	0.315	9.7, 5.9	9.0, 7.2	0.743	0.762	10.3, 11.9	8.7, 5.4	0.085	0.099
Neutrophils (%)	51.9, 28.9	42.4, 16.6	**0.016**	**0.003**	42.5, 17.4	42.9, 18.4	0.770	0.879	43.2, 20.7	42.0, 18.0	0.109	0.200
Lymphocytes (%)	28.8, 26.5	38.2, 18.1	**0.029**	**0.025**	40.5, 16.9	36.5, 21.6	0.707	0.982	35.2, 21.0	37.8, 19.3	0.176	0.123
NLR	1.1, 2.8	1.1, 1.0	0.461	0.741	1.2, 1.0	1.1, 1.5	0.587	0.674	1.2, 1.5	1.1, 1.2	0.308	0.389
Basophils (%)	0.8, 0.6	0.6, 0.6	**0.025**	0.135	0.6, 0.6	0.7, 0.6	0.335	0.487	0.7, 0.6	0.6,0.6	0.393	0.692
Eosinophils (%)	1.5, 1.1	1.9, 2.2	0.083	0.300	1.9, 1.9	1.8, 2.2	0.804	0.807	1.6, 1.8	2.0, 2.2	0.260	0.364
Monocytes (×10^9^)	4.9, 3.2	4.6, 2.4	0.387	0.562	5.2, 2.4	4.4, 2.2	**0.016**	0.991	5.1, 2.7	3.9, 2.3	**0.016**	0.587
Immature granulocytes (%)	0.9, 1.0	1.0, 0.7	0.857	0.940	1.2, 1.0	0.9, 0.6	0.317	0.146	1.0, 0.7	0.9, 0.8	0.470	0.461
PDW (%)	9.5, 2.3	9.6, 2.1	0.961	0.984	9.8, 1.6	9.5, 2.5	0.413	0.574	9.7, 1.7	9.4, 2.5	0.503	0.540
Total bilirubin (mg/dL)	6.1, 1.4	6.6, 2.5	0.125	0.451	6.7, 2.3	6.4, 2.2	0.306	0.620	6.4, 2.1	6.5, 2.3	0.796	0.874

*BDNF*, *brain-derived neurotrophic factor*; MCH, mean corpuscular hemoglobin; MCV, mean corpuscular volume; NLR, neutrophil/lymphocyte ratio; *p*, *p*-value; PDW, platelet distribution width; *p**, *p*-value adjusted for gestational age and the number of red blood cell transfusions; ROP, retinopathy of prematurity. *p*-values less than 0.05 are in bold. Mann‒Whitney test and multivariable logistic regression analysis.

**Table 6 ijms-26-00898-t006:** Relationships of the ***EPO*** polymorphism with hematological and biochemical parameters: (**a**) in the group of infants who did not develop ROP; (**b**) in the group of infants who developed ROP.

**(a) *EPO* (rs507392) Polymorphism in the No ROP Group**												
**Median, Interquartile Range**	**AA**	**Allele G**	** *p* **	** *p** **	**GG**	**Allele A**	** *p* **	** *p** **	**AA + GG**	**GA**	** *p* **	** *p** **
Erythrocytes (×10^12^/L)	4.3, 0.9	4.3, 0.8	0.908	0.728	4.4, 0.8	4.2, 0.9	0.907	0.543	4.3, 0.8	4.2, 0.9	0.847	0.972
MCV (fl)	106.9, 8.5	107.7, 10.2	0.803	0.884	107.1, 11.2	107.3, 9.7	0.633	0.430	107.0, 9.0	108.0, 10.3	0.566	0.744
MCH (pg)	38.1, 2.7	37.4, 4.0	0.226	0.220	36.8, 4.5	37.7, 3.2	0.307	0.129	38.0, 2.9	37.4, 3.8	0.634	0.824
Reticulocytes (%)	9.2, 4.4	6.5, 4.0	0.279	0.671	6.8, 3.3	6.7, 5.4	0.884	0.875	7.0, 3.2	5.5, 1.5	0.216	0.575
Neutrophiles (%)	42.7, 14.9	40.7, 16.4	0.445	0.588	36.9, 19.4	42.0, 15.4	**0.041**	**0.017**	41.4, 14.8	41.5, 15.9	0.520	0.369
LMR	3.4, 2.9	3.3, 2.3	0.829	0.755	4.0, 2.4	3.3, 2.7	0.486	0.247	3.5, 2.7	3.3, 2.4	0.487	0.279
Basophils (%)	0.5, 0.4	0.5, 0.4	0.531	0.744	0.5, 0.5	0.5, 0.4	0.296	0.319	0.5, 0.4	0.5, 0.4	0.186	0.330
Eosinophils (%)	2.3, 2.5	2.0, 2.4	0.063	0.300	1.6, 1.6	2.2, 2.5	**0.020**	0.168	2.1, 2.5	2.1, 2.5	0.574	0.925
Granulocytes (%)	1.2, 1.7	1.2, 0.7	0.961	0.823	1.0, 0.6	1.3, 1.3	0.256	0.348	1.2, 1.3	1.4, 1.0	0.555	0.807
Plateletcrit (%)	0.2, 0.1	0.3, 0.1	0.742	0.666	0.32, 0.1	0.24, 0.1	0.635	0.310	0.3, 0.1	0.3, 0.1	0.487	0.480
**(b) *EPO* (rs507392) Polymorphism in the ROP Group**												
**Median, Interquartile Range**	**AA**	**Allele G**	** *p* **	** *p** **	**GG**	**Allele A**	** *p* **	** *p** **	**AA + GG**	**GA**	** *p* **	** *p** **
Erythrocytes (×10^12^/L)	3.8, 0.8	4.0, 0.8	**0.024**	**0.043**	4.3, 0.8	4.0, 0.8	0.194	0.434	3.9, 0.8	4.0, 0.7	0.201	0.163
MCV (fl)	109.8, 10.9	107.7, 11.8	**0.044**	0.060	106.6, 18.3	108.7, 11.0	0.695	0.404	109.5, 12.7	108.0, 13.1	**0.029**	0.021
MCH (pg)	37.8, 2.9	36.9, 4.1	**0.017**	**0.023**	37.6, 5.5	37.1, 3.8	0.720	0.443	37.7, 3.3	36.7, 4.1	**0.012**	0.008
Reticulocytes (%)	10.4, 6.1	5.6, 3.1	**0.018**	0.994	6.3, 1.2	7.4, 6.5	0.758	0.582	8.1, 6.8	4.9, 3.9	**0.048**	0.171
Neutrophils (%)	42.2, 14.5	42.9, 19.8	0.523	0.643	42.2, 16.8	42.8, 18.2	0.700	0.745	42.2, 15.0	44.4, 21.8	0.386	0.600
LMR	3.0, 2.2	3.1, 3.1	0.854	0.792	5.0, 4.1	2.9, 2.3	**0.029**	**0.008**	3.3, 2.8	2.8, 2.5	0.118	0.079
Basophils (%)	0.6, 0.7	0.7, 0.6	0.394	0.364	0.9, 0.5	0.7, 0.6	**0.040**	0.543	0.7, 0.7	0.7, 0.5	0.564	0.629
Eosinophils (%)	1.9, 2.1	1.7, 2.0	0.476	0.463	2.3, 2.5	1.8, 1.9	0.485	0.119	2.0, 2.3	1.6, 1.9	0.255	0.288
Granulocytes (%)	1.1, 0.9	0.8, 0.6	0121	0.144	1.3, 0.6	0.9, 0.6	0.412	0.681	1.1, 0.8	0.7, 0.5	**0.029**	0.093
Plateletcrit (%)	0.2, 0.1	0.2, 0.1	0.264	0.641	0.15, 0.1	0.21, 0.1	**0.021**	**0.020**	0.2, 0.1	0.2, 0.1	0.685	0.489

*EPO*, *erythropoietin*; LMR, lymphocyte/monocyte ratio; MCH, mean corpuscular hemoglobin; MCV, mean corpuscular volume; *p*, *p*-value; *p**, *p*-value adjusted for gestational age and the number of red blood cell transfusions; ROP, retinopathy of prematurity. *p*-values less than 0.05 are in bold. Mann‒Whitney test and multivariable logistic regression analysis.

**Table 7 ijms-26-00898-t007:** Epistatic relationships between genetic polymorphisms and ROP.

Epistatic Relationships	No ROP *N* (%)	ROP *N* (%)	OR [CI 95%]	*p*	*p**
***NGF*** (GG) + ***EPO*** (GG)	9 (50.0)	9 (50.0)	n.a.	0.463	0.666
Other genotypes of ***NGF*** + ***EPO***	224 (60.2)	148 (39.8)			
***NGF*** (GG) + ***BDNF*** (Allele G)	61 (50.0)	61 (50.0)	1.79 [1.16–2.77]	**0.010**	0.069
Other genotypes of ***NGF*** +***BDNF***	174 (64.2)	97 (35.8)			
***NGF*** (GG) +***TH*** (Allele C)	51 (51.5)	48 (48.5)	n.a.	0.075	0.230
Other genotypes of ***NGF*** + ***TH***	176 (62.0)	108 (38.0)			
***EPO*** (GG) +***BDNF*** (Allele G)	23 (57.3)	17 (42.5)	n.a.	0.737	0.875
Other genotypes of ***EPO*** +***BDNF***	211 (60.1)	140 (39.9)			
***EPO*** (GG) + ***TH*** *(Allele C)*	19 (52.8)	17 (47.2)	n.a.	0.476	0.292
Other genotypes of ***EPO*** +***TH***	209 (60.1)	139 (39.9)			
***BDNF*** (Allele G) +***TH*** (Allele C)	105 (55.9)	83 (44.1)	n.a.	0.178	0.458
Other genotypes of ***BDNF*** +***TH***	123 (62.8)	73 (37.2)			
***BDNF*** (Allele G) +***EPO***(GG) + ***NGF*** (GG)	8 (50.0)	8 (50.0)	n.a.	0.444	0.672
Other genotypes of ***BDNF*** +***EPO** + **NGF***	225 (60.2)	149 (39.8)			
***NGF*** (GG)+***EPO***(GG) +***TH*** (Allele C)	8 (53.3)	7 (46.7)	n.a.	0.790	0.965
Other genotypes of ***NGF*** + ***EPO*** +***TH***	219 (59.5)	149 (40.5)			
***NGF*** (GG) + ***BDNF*** (Allele G) +***TH*** (Allele C)	38 (41.3)	54 (58.7)	2.38 [1.33–4.25]	**<0.001**	**0.003**
Other genotypes of ***NGF*** +***BDNF***+***TH***	190 (65.1)	102 (34.9)			
***EPO*** (GG) + ***BDNF*** (Allele G) +***TH*** (Allele C)	16 (57.1)	12 (42.9)	n.a.	0.843	0.639
Other genotypes of ***EPO*** +***BDNF***+ ***TH***	212 (59.6)	144 (40.4)			
***NGF*** (GG) +***EPO***(GG) +***BDNF***(Allele G) +***TH*** (Allele C)	221 (59.4)	151 (40.6)	n.a.	0.764	0.778
Other genotypes of ***NGF*** + ***EPO*** +***BDNF***+ ***TH***	6 (54.5)	5 (45.5)			

*BDNF*, *brain-derived neurotrophic factor*; *EPO*, *erythropoietin*; n.a., not applicable; *NGF*, *nerve growth factor*; OR, odds ratio; *p*, *p*-value; *p**, *p*-value adjusted for gestational age and the number of red blood cell transfusions; ROP, retinopathy of prematurity; *TH*, *tyrosine hydroxylase*. *p*-values less than 0.05 are in bold. Multivariable logistic regression analysis.

**Table 8 ijms-26-00898-t008:** Epistatic relationships between polymorphisms of ***NGF*** and ***BDNF***, ***NGF*** and ***TH***, and ***BDNF*** and ***TH*** with hematological parameters: (**a**) in the group of infants who did not develop ROP; (**b**) in the group of infants who developed ROP.

**(a) Epistatic Relationships in the Group Without ROP**												
***N*; Median, Interquartile Range**	***NGF* (GG) + *BDNF* (Allele G)**	**Other Genotypes of *NGF* + *BDNF***	** *p* **	** *p** **	***NGF* (GG) + *TH* (Allele C)**	**Other Genotypes of *NGF* + *TH***	** *p* **	** *p** **	***BDNF* (Allele G) + *TH* (Allele C)**	**Other Genotypes of *BDNF* + *TH***	** *p* **	** *p** **
Lymphocytes (×10^9^/L)	38; 4.0, 2.0	94; 4.0, 1.8	0.815	0.974	32; 4.1, 2.0	94; 4.0, 2.0	0.901	0.917	94; 3.8, 1.6	33; 4.5, 2.3	**0.035**	0.756
Lymphocytes (%)	48; 44.9, 16.7	139; 40.0, 16.4	0.053	0.122	55; 45.2, 16.7	124; 40.0, 15.3	0.053	0.096	84; 40.8, 16.6	96; 40.6, 17.4	0.938	0.685
Monocytes (×10^9^/L)	35; 1.1, 0.9	82; 1.2, 0.8	0.666	0.701	31; 1.13, 0.6	82; 1.14, 0.9	0.688	0.689	84; 1.1, 0.8	30; 1.2, 0.9	0.215	0.256
Monocytes (%)	11; 4.8, 2.3	24; 4.9, 1.9	0.201	0.337	12; 4.5, 2.2	23; 4.9, 1.9	0.2	0.301	15; 4.9, 1.3	20; 4.6, 1.9	0.400	0.655
Immature granulocytes (%)	12; 0.8, 0.5	39; 1.5, 1.3	**0.007**	**0.009**	13; 1.0, 1.8	35; 1.3, 0.8	0.634	0.699	34; 1.1, 1.0	15; 1.6, 1.6	**0.028**	0.083
**(b) Epistatic Relationships in the Group with ROP**												
***N*; Median, Interquartile Range**	***NGF* (GG) + *BDNF* (Allele G)**	**Other Genotypes of *NGF* + *BDNF***	** *p* **	** *p** **	***NGF* (GG) + *TH* (Allele C)**	**Other Genotypes of *NGF* + *TH***	** *p* **	** *p** **	***BDNF* (Allele G) + *TH* (Allele C)**	**Other Genotypes of *BDNF* + *TH***	** *p* **	** *p** **
Lymphocytes (×10^9^/L)	43; 3.0, 1.2	68; 3.7, 2.1	0.137	0.459	36; 2.9, 1.2	73; 3.7, 1.9	**0.017**	**0.035**	74; 3.5, 1.8	35; 3.6, 1.4	0.736	0.779
Lymphocytes (%)	52; 38.5, 19.9	89; 36.1, 21.2	0.284	0.310	49; 36.7, 20.9	90; 37.6, 19.7	1.000	1.000	74; 40.8, 18.4	65; 34.6, 23.0	**0.040**	**0.026**
Monocytes (×10^9^/L)	41; 1.1, 0.8	66; 1.3, 1.2	0.136	0.197	34; 1.0, 1.0	71; 1.2, 1.1	**0.025**	**0.028**	71; 1.1, 0.8	34; 1.3, 1.5	0.468	0.502
Monocytes (%)	27; 4.6, 2.4	41; 4.7, 2.5	0.995	0.999	19; 4.2, 2.2	49; 4.7, 2.5	0.667	0.769	26; 4.9, 2.8	42; 4.1, 2.0	**0.038**	0.220
Immature granulocytes (%)	8; 1.2, 0.5	13; 0.9, 0.5	0.506	0.681	9; 1.0, 0.5	12; 0.9, 1.0	0.393	0.501	16; 0.9, 0.6	5; 0.9, 1.1,	0.563	0.627

*BDNF*, *brain-derived neurotrophic factor*; *NGF*, *nerve growth factor*; *p*, *p*-value; *p**, *p*-value adjusted for gestational age and the number of red blood cell transfusions; ROP, retinopathy of prematurity; *TH*, *tyrosine hydroxylase*; *p*-values less than 0.05 are in bold.

**Table 9 ijms-26-00898-t009:** Epistatic relationships between polymorphisms of *EPO* and *NGF*, *EPO* and *TH*, and *EPO* and *BDNF* with hematological parameters: (**a**) in the group of infants who did not develop ROP; (**b**) in the group of infants who developed ROP.

**(a) Epistatic Relationships in the Group Without ROP**												
***N*; Median, Interquartile Range**	** *EPO* ** **(GG) + *NGF* (GG)**	**Other Genotypes of *NGF* + *EPO***	** *p* **	** *p** **	***EPO* (GG) + *TH* (Allele C)**	**Other Genotypes of EPO + TH**	** *p* **	** *p** **	***EPO* (GG) + *BDNF* (Allele G)**	**Other Genotypes of EPO + BDNF**	** *p* **	** *p** **
MCH	8; 36.4, 7.6	172; 37.7, 3.2	0.349	0.772	17; 36.1, 3.9	158; 37.8, 3.2	**0.017**	**0.006**	21; 36.8, 4.5	160; 37.7, 3.2	0.363	0.558
Basophils (%)	9; 0.5, 0.3	177; 0.5, 0.4	0.570	0.893	17; 0.53, 0.5	164; 0.53, 0.4	0.693	0.577	21; 0.5, 0.5	166; 0.5, 0.4	0.443	0.610
Eosinophils (×10^3^/L)	7; 0.2, 0.1	122; 0.2, 0.3	0.658	0.725	11; 0.1, 0.1	116; 0.2, 0.2	**0.023**	0.504	14; 0.1, 0.1	116; 0.2, 0.3	**0.007**	**0.032**
Monocytes (×10^9^/L)	7; 1.2, 1.1	108; 1.1, 0.8	0.317	0.400	10; 1.1, 0.7	104; 1.1, 0.8	0.924	0.993	13; 1.1, 0.7	103; 1.2, 0.8	0.700	0.354
LMR	7; 2.8, 1.8	108; 3.4, 2.7	0.368	0.394	10; 4.0, 3.2	104; 3.3, 2.6	0.631	0.713	13; 4.3, 3.9	103; 3.3, 2.7	0.304	0.111
Platelets (×10^9^/L)	8; 168.1, 84.9	175; 226.0, 108.0	**0.022**	**0.012**	17; 218.0, 93.7	162; 223.7, 110.7	0.813	0.886	21; 212.0, 93.7	163; 225.0, 111.0	0.612	0.346
Platelecrit (%)	5; 0.2, 0.1	84; 0.3, 0.1	0.154	0.090	11; 0.27, 0.1	78; 0.24, 0.1	0.658	0.811	14; 0.3, 0.1	76; 0.2, 0.1	0.885	0.423
PDW (%)	5; 8.9, 2.6	102; 10.2, 1.4	0.268	0.503	10; 9.1, 2.0	97; 10.3, 1.4	**0.046**	**0.025**	14; 10.1, 2.3	94; 10.2, 1.3	0.819	0.874
**(b) Epistatic Relationships in the Group with ROP**												
***N*; Median, Interquartile Range**	** *EPO* ** **(GG) + *NGF* (GG)**	**Other Genotypes of *NGF* + *EPO***	** *p* **	** *p** **	***EPO* (GG) + *TH* (Allele C)**	**Other Genotypes of *EPO* + *TH***	** *p* **	** *p** **	***EPO* (GG) + *BDNF* (Allele G)**	**Other Genotypes of *EPO* + *BDNF***	** *p* **	** *p** **
MCH	8; 38,2, 8.6	132; 37.0, 3.8	0.165	0.412	13; 37.7, 5.4	126; 37.1, 3.9	0.662	0.569	14; 37.6, 5.4	126; 37.1, 3.8	0.541	0.812
Basophils (%)	8; 0.7, 0.5	125; 0.7, 0.6	0.677	0.910	12; 1.0, 0.5	120; 0.6, 0.6	**0.011**	0.351	14; 0.8, 0.5	119; 0.7, 0.6	0.134	0.209
Eosinophils (×10^3^/L)	5; 0.1, 0.1	100; 0.2, 0.2	0.102	0.193	9; 0.1, 0.1	95; 0.2, 0.2	0.422	0.895	9; 0.1, 0.2	96; 0.2, 0.2	0.464	0.888
Monocytes (×10^9^/L)	5; 0.7, 1.0	101; 1.2, 1.0	0.192	0.611	7; 0.4, 0.4	98; 1.2, 1.0	**0.013**	0.963	9; 0.5, 1.1	97; 1.2, 1.0	**0.045**	0.948
LMR	5; 3.5, 8.7	101; 3.0, 2.6	0.392	0.758	7; 5.3, 10.6	98; 2.9, 2.4	**0.004**	0.051	9; 5.1, 7.7	97; 3.0, 2.3	**0.017**	**0.004**
Platelets (×10^9^/L)	8; 188.5, 110.1	133; 206.7, 123.4	**0.047**	0.060	11; 128.5, 146.3	129; 205.0, 120.4	**0.006**	0.065	14; 188.5, 138.9	127; 206.7, 123.3	**0.011**	**0.009**
Platelecrit (%)	6; 0.16, 0.04	89; 0.21, 0.1	**0.041**	0.066	6; 0.16, 0.1	88; 0.21, 0.1	**0.008**	0.431	10; 0.15, 0.1	85; 0.21, 0.1	**0.010**	**0.012**
PDW (%)	7; 9.3, 3.4	96; 9.6, 2.1	0.508	0.587	10; 8.9, 2.0	92; 9.7, 2.0	0.235	0.787	11; 8.8, 3.0	92; 9.7, 2.0	0.290	0.249

*BDNF*, *brain-derived neurotrophic factor*; *EPO*, *erythropoietin*; LMR, lymphocyte/monocyte ratio; MCH, mean corpuscular hemoglobin; *NGF*, *nerve growth factor*; *p*, *p*-value; *p**, *p*-value adjusted for gestational age and the number of red blood cell transfusions; PDW, platelet distribution width; ROP, retinopathy of prematurity; *TH*, *tyrosine hydroxylase*. *p*-values less than 0.05 are in bold.

**Table 10 ijms-26-00898-t010:** Epistatic relationships between polymorphism of ***NGF***, ***BDNF***, and ***EPO***, ***EPO***, ***BDNF***, and ***TH***, and ***NGF***, ***BDNF***, and ***TH*** with hematological parameters: (**a**) in the group of infants who did not develop ROP; (**b**) in the group of infants who developed ROP.

**(a) Epistatic Relationships in the Group Without ROP**												
***N*; Median, Interquartile Range**	***NGF* (GG) + *BDNF* (Allele G) + *EPO* (GG)**	**Other Genotypes of *NGF* + *BDNF* + *EPO***	** *p* **	** *p** **	***EPO* (GG) + *BDNF* (Allele G) + *TH* (Allele C)**	**Other Genotypes of *EPO* + *BDNF* + *TH***	** *p* **	** *p** **	***NGF* (GG) + *BDNF* (Allele G) + *TH* (Allele C)**	**Other Genotypes of *NGF* + *BDNF* + *TH***	** *p* **	** *p** **
MCH	7; 38.2, 8.4	173;37.7, 3.2	0.660	0.906	14; 35.8, 4.7	161; 37.7, 3.1	**0.048**	**0.013**	41; 38.1, 3.8	134; 37.5, 3.2	0.300	0.571
Eosinophils (×10^3^/L)	6; 0.1, 0.1	123; 0.2, 0.3	0.247	0.444	8; 0.1, 0.08	119; 0.2, 0.3	**0.022**	0.069	32; 0.3, 0.3	95; 0.2, 0.3	0.311	0.297
Immature granulocytes (%)	2; 0.8, n.a.	47; 1.2, 1.1	0.189	0.312	3; 1.0, n.a.	46; 1.3, 1.3	0.287	0.346	5; 0.6, 0.5	16; 1.4, 1.3	**0.004**	**0.025**
Platelecrit (%)	5; 0.2, 0.1	84; 0.3, 0.1	0.154	0.090	11; 0.27, 0.1	78; 0.24, 0.1	0.658	0.804	19; 0.3, 0.1	70; 0.3, 0.1	0.996	0.999
**(b) Epistatic Relationships in the Group with ROP**												
***N*; Median, Interquartile Range**	***NGF* (GG) + *BDNF* (Allele G) + *EPO* (GG)**	**Other Genotypes of NGF + *BDNF* + *EPO***	** *p* **	** *p** **	***EPO* (GG) + *BDNF* (Allele G) + *TH* (Allele C)**	**Other Genotypes of EPO + *BDNF* + *TH***	** *p* **	** *p** **	***NGF* (GG) + *BDNF* (*Allele* G) + *TH* (Allele C)**	**Other Genotypes of NGF + *BDNF* + *TH***	** *p* **	** *p** **
MCH	7; 37.7, 6.9	133; 37.1, 3.8	0.411	0.769	9; 37.7, 4.4	130; 37.1, 3.9	0.419	0.390	36; 36.8, 5,0	103; 37.4, 3.7	0.192	0.213
Eosinophils (×10^3^/L)	4; 0.1, 0.1	101; 0.2, 0.2	0.172	0.227	14; 35.8, 4.7	161; 37.7, 3.1	**0.048**	**0.013**	29; 0.2, 0.2	75; 0.2, 0.2	0.919	0.954
Immature granulocytes (%)	3; 1.3, n.a.	18; 0.9, 0.6	0.365	0.610	2; 1.0, n.a.	19; 0.9, 0.7	0.905	0.924	5; 1.1, 1.2	16; 0.9, 0.6	0.501	0.332
Platelecrit (%)	6; 0.16, 0.04	89; 0.21, 0.1	**0.041**	0.066	6; 0.16, 0.1	88; 0.21, 0.1	**0.008**	0.062	25; 0.2, 0.1	69; 0.2, 0.1	0.294	0.521

*BDNF*, *brain-derived neurotrophic factor*; *EPO*, *erythropoietin*; MCH, mean corpuscular hemoglobin; n.a., not applicable; *NGF*, *nerve growth factor*; *p*, *p*-value; *p**, *p*-value adjusted for gestational age and the number of red blood cell transfusions; ROP, retinopathy of prematurity; *TH*, *tyrosine hydroxylase*. *p*-values less than 0.05 are in bold.

## Data Availability

The datasets used and analyzed during this study are available from the corresponding author upon reasonable request.

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
