# Peer review of "Influence of Functional Variations in Genes of Neurotrophins and Neurotransmitter Systems on the Development of Retinopathy of Prematurity"

_ijms, 2025, doi:10.3390/ijms26030898_

Round 1
Reviewer 1 Report
Comments and Suggestions for Authors
This is an interesting and very well-written manuscript that performs a comparative biochemical, hematological and genotypic analysis of infants affected or not with ROP. I have a few commentaries so that the manuscript can be improved:
In the introduction, the authors state “Considering the roles of NGF, BDNF, TH, and EPO in retinal neurodevelopment”. Please, include a reference(s) to support this statement.
Please discuss the importance of hyperglycemia, which can lead to retinopathy, in ROP patients.
Please discuss the clinical implications of at least some hematological findings with correlation to the genotypes. For example, a higher ratio of neutrophils/lymphocytes and TH polymorphisms. Please discuss how neutrophils/ lymphocytes (for example, Tregs) and eosinophils can participate in ROP, especially neutrophil-mediated tissue damage (also for the other genotypes of BDNF and NGF).
Please add references to the statement in lines 491-492.
Please revise the English grammar. For example, substitute “associated to” to “associated with”.
Is there any study demonstrating how the GG polymorphism of NGF affects the levels of this growth factor? What about the other genotypes and the levels of the other growth factors?
The authors mentioned more than once how TH might regulate thermogenesis. Please discuss how thermogenesis and brown fat tissue can influence ROP development.
The number of control infants can be reduced, especially to restrain the significant differences between ROP and healthy infants in relation to gestational age and birth weight. As both can influence the health of the infant, these differences should be suppressed so the biochemical and genotypic analysis performed can be considered relevant in an independent manner from GA and birth weight.
Author Response
Comment 1: In the introduction, the authors state “Considering the roles of NGF, BDNF, TH, and EPO in retinal neurodevelopment”. Please, include a reference(s) to support this statement.
Answer: Thank you for your valuable comment. We have included the appropriate references to support this statement in the revised manuscript (line 115). These references address the roles of NGF, BDNF, TH, and EPO in retinal neurodevelopment, as well as in vascularization, homeostasis, and stress regulation.
Comment 2: Please discuss the importance of hyperglycemia, which can lead to retinopathy, in ROP patients.
Answer: Thank you for your insightful comment. We have addressed the relevance of hyperglycemia in the context of ROP in the revised manuscript. A new section has been added to the Introduction (lines 103-113) to discuss how hyperglycemia, as a common metabolic consequence of stress responses in preterm infants, may influence retinal neurovascular regulation.
Hyperglycemia has been associated with increased oxidative stress and systemic inflammation, both of which are critical contributors to neurovascular damage and potentially exacerbate the pathogenesis of ROP. Furthermore, we highlight the role of stress-induced hormonal activation, such as the hypothalamic-pituitary-adrenal (HPA) axis, in driving hyperglycemia. While the literature presents conflicting findings regarding its direct contribution to ROP risk or severity, we acknowledge its potential impact and have cited relevant studies to support this discussion.
In our study, however, no direct association was observed between glycemic parameters and the polymorphisms analyzed in preterm infants with or without ROP, which is why hyperglycemia was not a focus in the Results or Discussion sections. Nonetheless, its potential role as a modifiable metabolic factor remains an important consideration in the broader context of ROP pathophysiology.
Comment 3: Please discuss the clinical implications of at least some hematological findings with correlation to the genotypes. For example, a higher ratio of neutrophils/lymphocytes and TH polymorphisms. Please discuss how neutrophils/ lymphocytes (for example, Tregs) and eosinophils can participate in ROP, especially neutrophil-mediated tissue damage (also for the other genotypes of BDNF and NGF).
Answer: We thank the reviewer for their valuable suggestion to discuss the clinical implications of hematological findings in relation to the genotypes studied, specifically focusing on the neutrophil-to-lymphocyte ratio (NLR), Tregs, eosinophils, and neutrophil-mediated tissue damage. We have carefully addressed these points and integrated the following updates into the manuscript:
- Tyrosine Hydroxylase (TH) Polymorphism: We discuss the association between the TH rs10770141 polymorphism and elevated NLR, highlighting how the allele C is linked to increased neutrophil counts and the potential for neutrophil-mediated retinal damage through pro-inflammatory mechanisms. The implications of reduced lymphocyte counts, particularly Tregs, in modulating inflammation are also adressed. This discussion is included in lines 481–493 and 497–499.
- Brain-Derived Neurotrophic Factor (BDNF) Polymorphism: The discussion has been revised and expanded to include the relationship between the BDNF rs7934165 polymorphism and its effects on neutrophil and lymphocyte percentages, as well as the potential role of allele A in diminishing BDNF-mediated inflammatory regulation. Previous content has been refined or replaced to ensure clarity and incorporate these updates. These changes can be found in lines 517–536 of the revised manuscript.
- Nerve Growth Factor (NGF) Polymorphism: While no direct association between the NGF polymorphism and NLR was observed, we expanded the discussion to explore NGF’s dual roles in vascular repair and neuroprotection, alongside its pro-angiogenic and inflammatory modulation under hypoxic stress. This interplay may influence ROP development. Additionally, we discuss NGF’s role in T cell recruitment and immune modulation, highlighting the need for further investigation into its contributions to ROP. These updates can be found in lines 412–422.
These revisions aim to provide a clearer understanding of the interplay between the studied genotypes, hematological parameters, and their potential roles in ROP pathogenesis. We hope these changes meet the reviewer’s expectations and enhance the manuscript's scientific quality.
Comment 4: Please add references to the statement in lines 491-492.
Answer: Thank you for your comment. We have added references to support the statement regarding the likely role of impaired inflammatory control in the observed differences in granulocyte percentages among the genotype combinations discussed. The revised text now cites studies highlighting the role of NGF, BDNF, and TH in regulating inflammatory responses (references added: Minnone et al., 2017 [43]; Sochal et al., 2022 [75]; Channer et al., 2023 [74]). While these references do not directly address the specific polymorphisms studied in our cohort, they provide evidence for the broader involvement of NGF, BDNF, and TH in modulating inflammation, which aligns with the proposed explanation.
Comment 5: Please revise the English grammar. For example, substitute “associated to” to “associated with”.
Answer: Thank you for your observation. We have carefully reviewed and revised the grammar throughout the manuscript. Specifically, the substitution of “associated to” with “associated with” has been corrected in lines 209 and 513, as requested.
Comment 6: Is there any study demonstrating how the GG polymorphism of NGF affects the levels of this growth factor? What about the other genotypes and the levels of the other growth factors?
Answer: Thank you for your question. Regarding the polymorphisms studied:
- NGF (rs6330): Zakharyan et al., 2014 [45] demonstrated a positive association between the presence of the minor allele (A) and decreased plasma levels of NGF protein. This finding suggests that individuals with the minor allele (allele A) may exhibit lower NGF levels compared to those with the genotype GG. We have incorporated this information into the manuscript (lines 375–378) to better contextualize the potential impact of the rs6330 polymorphism on NGF levels.
- BDNF (rs7934165): Regarding this polymorphism, the manuscript originally included the statement: "The minor allele (A) of the BDNF rs7934165 polymorphism is associated with increased methylation at several CpG sites, reducing gene expression." To enhance clarity and provide additional context, this sentence has been refined in the revised manuscript (lines 503–505). We have also reinforced this explanation by adding: "This epigenetic modification may decrease BDNF protein availability, which could impact its neurotrophic and immunomodulatory functions" (lines 506–507). Additionally, we highlighted that "higher BDNF expression associated with the allele G may confer anti-inflammatory effects" (lines 518–521). These additions seek to ensure a more explicit connection between the methylation changes, reduced gene expression, and their functional consequences.
- TH (rs10770141): Lee et al. (2016) investigated this polymorphism in the promoter region of the TH gene and reported that the allele T is associated with increased catecholamine secretion, including norepinephrine. This finding suggests that the polymorphism may influence the activity of the sympathetic nervous system, potentially impacting stress and homeostasis regulation—factors relevant to conditions such as ROP. This information was already included in the original manuscript (lines 449–451), and no further changes were made.
- EPO (rs507392): While studies, such as those by Kaur et al. (2021) and Sesti et al. (2022) (references 71 and 67), have explored the association of EPO polymorphisms with disease outcomes like diabetic retinopathy, no direct evidence has been identified linking the rs507392 polymorphism to serum levels or gene expression of EPO. Further research would be necessary to elucidate these potential relationships. Accordingly, no changes were made to the manuscript regarding this polymorphism.
These findings provide insights into how the genetic variants analyzed in our study might influence the levels or activity of their respective factors. While direct measurements of serum levels were not universally available, the evidence from these studies highlights potential mechanistic links between the polymorphisms and the biological pathways they regulate.
Comment 7: The authors mentioned more than once how TH might regulate thermogenesis. Please discuss how thermogenesis and brown fat tissue can influence ROP development.
Answer: Thank you for your valuable comment regarding the role of brown adipose tissue (BAT) and thermogenesis in retinopathy of prematurity (ROP). We have addressed this by elaborating on the role of tyrosine hydroxylase (TH) in regulating BAT activity through catecholamine synthesis. BAT sustains thermogenesis via uncoupling protein 1 (UCP1)-mediated oxidative phosphorylation, a process requiring significant oxygen consumption. This connection underscores BAT’s potential role in systemic oxygen homeostasis. Reduced BAT activity may exacerbate hyperoxia in the first phase of ROP, while increased thermogenic demand could contribute to hypoxia during the second phase.
Additionally, we expanded the discussion to include the potential downstream effects of reduced BAT formation and oxygen consumption on erythropoietin (EPO) production.
These insights are now included in the Introduction and Discussion sections (lines 81 to 91, 442 to 448, and 579 to 581) to highlight how TH (rs10770141) polymorphism and BAT regulation may influence ROP development. Further research is warranted to clarify these mechanisms.
We hope these revisions align with the reviewer’s request and improve the manuscript.
Comment 8: The number of control infants can be reduced, especially to restrain the significant differences between ROP and healthy infants in relation to gestational age and birth weight. As both can influence the health of the infant, these differences should be suppressed so the biochemical and genotypic analysis performed can be considered relevant in an independent manner from GA and birth weight.
Answer: Thank you for raising this important point. We recognize that gestational age (GA) and birth weight (BW) are critical factors influencing the health of preterm infants and may affect the hematological and biochemical parameters analyzed in this study. Below, we outline the rationale behind our approach and the adjustments made in the study:
- Role of GA and BW in ROP: While BW has been identified as an important risk factor for ROP in several studies, our logistic regression analysis identified GA and the number of red blood cell transfusions as the most significant risk factors (described in the manuscript, lines 753 to 759), with BW losing significance in the multivariate model. Based on these findings, we adjusted our analyses for GA and the number of transfusions, as these factors were deemed the most relevant confounders in the context of this cohort and the parameters analyzed (lines 141 to 144).
- Rationale for not adjusting for BW: Although BW is an important marker of neonatal health, its influence in this cohort appears to be largely mediated by GA. Given the multifactorial nature of ROP, it is challenging to account for all potential risk factors. By prioritizing adjustments for GA and transfusions, we focused on the most statistically and clinically significant variables, ensuring relevance to the specific context of this study.
- Number of Controls: The number of control infants in our study reflects the natural composition of the cohort, ensuring sufficient representation for the planned genetic and biochemical analyses. Reducing the number of controls might compromise the robustness of our findings and increase the likelihood of type II errors, particularly given the complexity of multifactorial conditions like ROP.
In summary, although BW is a recognized risk factor for ROP, in this cohort and for the parameters analyzed, adjustments for GA and the number of red blood cell transfusions were deemed the most appropriate to control for confounding effects. This approach reflects our effort to address key confounders while acknowledging the inherent complexity of studying multifactorial diseases like ROP.
Reviewer 2 Report
Comments and Suggestions for Authors
Dear Authors and Editors,
As a reviewer of the manuscript titled “Influence of Functional Variations in Genes of Neurotrophins and Neurotransmitter Systems on the Development of Retinopathy of Prematurity” by Mariza do Rosário Fevereiro-Martins, Ana Carolina Casinhas Santos, Carlos Alberto Matinho Marques-Neves, et al., I would like to commend the authors for the high quality of their work and for addressing the critical issue of retinopathy of prematurity (ROP).
ROP is a vasoproliferative disease that can lead to severe vision loss in premature infants and is a major cause of childhood blindness. The global pooled prevalence of ROP is 31.9%, with severe ROP affecting approximately 7.5% of premature infants. These statistics highlight the importance of ongoing research to better understand and address this condition.
This manuscript focuses on the role of nerve growth factor (NGF), brain-derived neurotrophic factor (BDNF), tyrosine hydroxylase (TH), and erythropoietin (EPO) genetic functional polymorphisms in the risk of developing ROP. The authors identified that the GG genotype of the NGF (rs6330) polymorphism may significantly increase the risk of ROP. Furthermore, the findings suggest that polymorphisms in genes associated with neurodevelopment, angiogenesis, homeostasis, neuroendocrine, and neuro-immune pathways may contribute to the etiopathogenesis of ROP. These insights hold the potential for enhancing ROP diagnostics in the future, enabling earlier detection and treatment to protect children's vision.
The authors conducted a multicenter study with well-defined inclusion criteria and employed advanced genotyping techniques, including the MicroChip DNA and iPlex MassARRAY® platform, which provide reliable and high-quality results. The manuscript is well-structured, presenting a comprehensive introduction and a thorough description of the methodology.
However, I would like to offer a few suggestions for improvement:
1)Introduction and Methods: Certain sections exhibit a relatively high percentage of text similarity with existing literature. To enhance originality and readability, I kindly recommend rephrasing or restructuring these sections while maintaining the scientific rigor.
2)Visualization: Including a schematic representation of the study design and key findings would significantly enhance the clarity and interpretability of the results. This would also aid readers in better understanding the study's implications.
3) Clinical Applications and Future Directions: Expanding on the clinical implications of these findings and proposing potential future research avenues, such as studies on gene-environment interactions or larger cohort analyses, would add depth to the discussion.
Overall, this study makes a valuable contribution to the field of neonatal and genetic research and provides novel insights into the genetic mechanisms underlying ROP. I recommend the manuscript for publication, contingent upon minor revisions to address the points mentioned above. These adjustments will further strengthen the impact and clarity of this important work.

Author Response
1)Introduction and Methods: Certain sections exhibit a relatively high percentage of text similarity with existing literature. To enhance originality and readability, I kindly recommend rephrasing or restructuring these sections while maintaining the scientific rigor.
Answer: Thank you for your valuable feedback regarding the text similarity in the Introduction and Methods sections. We appreciate your observation and have carefully reviewed these sections to ensure originality and enhance readability while maintaining the scientific rigor and accuracy of the content. We are confident that these updates align with your suggestions and improve the overall quality of the manuscript. Thank you again for your constructive comments.
2)Visualization: Including a schematic representation of the study design and key findings would significantly enhance the clarity and interpretability of the results. This would also aid readers in better understanding the study's implications.
Answer: We appreciate the reviewer’s suggestion to include a schematic representation of the study design and key findings to enhance clarity and interpretability. To address this, we have created a visual schematic summarizing the study design, key genetic polymorphisms analyzed, associated hematological findings, and their potential roles in the pathophysiology of ROP. This diagram highlights the flow from genetic variation to hematological biomarkers and their proposed implications for ROP development, providing a clear and concise overview for readers.
The schematic is included as Figure 1 in the revised manuscript and is referenced in lines 632–636 to facilitate understanding. We hope this addition meets the reviewer’s expectations and improves the manuscript’s accessibility and interpretability.
3) Clinical Applications and Future Directions: Expanding on the clinical implications of these findings and proposing potential future research avenues, such as studies on gene-environment interactions or larger cohort analyses, would add depth to the discussion.
Answer: We thank the reviewer for their valuable suggestion to expand on the clinical implications and propose future research avenues. We have addressed this by incorporating a new subsection, “3.6. Clinical Implications and Future Directions,” which discusses the potential clinical applications of our findings, including the development of genetic risk models and strategies for early intervention in preterm infants at risk of ROP. This section also outlines future research avenues, such as exploring gene-environment interactions, validating findings in larger and more diverse cohorts, and conducting longitudinal studies on neurodevelopmental outcomes.
To further enhance the organization and clarity of the manuscript following the inclusion of subsection 3.6, we have reorganized the content on the study's limitations into a distinct subsection, “3.7. Limitations of the Study.” While the limitations were already addressed in the original manuscript, presenting them as a separate subsection allows us to consolidate the discussion and better emphasize areas requiring caution and future investigation. To align with these changes and reinforce the key points, we have also revised the Conclusions (Section 5) to reflect the expanded clinical implications and future research directions.
These updates are detailed in lines 608–636 for 3.6, 637–645 for 3.7, and 778–788 for 5 (Conclusions) of the revised manuscript.
We hope these additions meet the reviewer’s expectations and enhance the depth and clarity of the discussion.
Reviewer 3 Report
Comments and Suggestions for Authors
Thank you for the opportunity to review the article entitled "Influence of Functional Variations in Genes of Neurotrophins and Neurotransmitter Systems on the Development of Retinopathy of Prematurity".
This study examines the impact of functional polymorphisms in nerve growth factor, brain-derived neurotrophic factor, tyrosine hydroxylase, and erythropoietin genes on the modulation of hematological and biochemical parameters during the first week of life, as well as their correlation with the development of retinopathy of prematurity. This is a challenging and intriguing problem; I commend the authors for addressing this assignment.
The Abstract and Introduction in a comprehensive and effective way introduce the interested reader to the issues discussed in the further parts of the manuscript. I have no reservations about these sections.
I find it incomprehensible that the methodological section is included in the final part of the manuscript - initially I suspected the authors of an unusual and unacceptable lack of this section in this type of work, which would disqualify the work in its entirety. I recommend positioning the methodological section appropriately within the publication, specifically directly following the introduction and the discussion of the study's purpose.
Apart from the above, my reservations concern the following elements:
1. In lines 540-541, the authors describe that the inclusion and exclusion criteria for the study are detailed in a previous publication; both documents should be regarded separately, allowing the reader to evaluate the technique without necessitating reference to another work. I recommend eliminating this element or clearly defining it within this manuscript.
2. In lines 564-571, the authors describe the laboratory parameters assessed - I suggest adding information about the center testing the samples and the biochemical analyzers used, as well as the appropriate reference value limits.
The presented results do not raise any major reservations.
The discussion is comprehensive and engaging for the audience. Appropriate references to other works were utilized, three of which are self-citations; yet, their inclusion does not provoke any significant concerns.
The presented conclusions accurately reflect the obtained results. I have no reservations about them.
The work does not bear the marks of plagiarism, either partial or complete.
In summary, the study possesses significant promise; yet, it necessitates revisions, particularly regarding its organization and methods.
Author Response
This study examines the impact of functional polymorphisms in nerve growth factor, brain-derived neurotrophic factor, tyrosine hydroxylase, and erythropoietin genes on the modulation of hematological and biochemical parameters during the first week of life, as well as their correlation with the development of retinopathy of prematurity. This is a challenging and intriguing problem; I commend the authors for addressing this assignment.
The Abstract and Introduction in a comprehensive and effective way introduce the interested reader to the issues discussed in the further parts of the manuscript. I have no reservations about these sections.
I find it incomprehensible that the methodological section is included in the final part of the manuscript - initially I suspected the authors of an unusual and unacceptable lack of this section in this type of work, which would disqualify the work in its entirety. I recommend positioning the methodological section appropriately within the publication, specifically directly following the introduction and the discussion of the study's purpose.
Answer: We thank the reviewer for their thoughtful feedback and for acknowledging the scientific value of our manuscript. We greatly appreciate the recognition of our efforts to address this challenging and important topic.
Regarding the placement of the Methods section, we would like to clarify that its current positioning adheres to the International Journal of Molecular Sciences guidelines. As outlined in the Instructions for Authors, the Methods section is typically placed after the Results and Discussion, aligning with the journal's standard structure to facilitate the logical flow of information.
We understand the reviewer's suggestion that placing the Methods section earlier could enhance clarity for some readers. However, altering this structure would require approval from the journal's editorial team, as it deviates from the established format. If such a change is recommended, we are fully prepared to adjust the manuscript accordingly.
Once again, we sincerely thank the reviewer for their thoughtful feedback.
Apart from the above, my reservations concern the following elements:
1.In lines 540-541, the authors describe that the inclusion and exclusion criteria for the study are detailed in a previous publication; both documents should be regarded separately, allowing the reader to evaluate the technique without necessitating reference to another work. I recommend eliminating this element or clearly defining it within this manuscript.
Answer: Thank you for your observation. We acknowledge that referring readers to another publication for key methodological details could be inconvenient. Upon review, we confirm that the inclusion and exclusion criteria are fully described within this manuscript (lines 661–669). The mention of the previous publication was intended only to acknowledge its prior reporting. To avoid redundancy and ensure clarity, we have removed the sentence referencing the previous publication.
- In lines 564-571, the authors describe the laboratory parameters assessed - I suggest adding information about the center testing the samples and the biochemical analyzers used, as well as the appropriate reference value limits.
Answer: Thank you for your valuable suggestion. The laboratory parameters were analyzed at the pathology services of the participating hospital centers, each employing standardized methods aligned with international best practices for laboratory testing. While specific brands of biochemical and hematological analyzers may vary slightly between centers, the following methodologies were consistently used:
- Hematological Parameters: Hemograms were performed using automated hematology analyzers, such as Sysmex®️ or Beckman Coulter®️ models, which utilize impedance and flow cytometry for cell counting, indices, and hemoglobin measurement.
- C-Reactive Protein (CRP): CRP was measured in plasma or serum samples using nephelometric and immunoturbidimetric assays on automated biochemistry analyzers (e.g., Roche Cobas®️, Abbott Architect®️, Olympus®️, Beckman-Coulter®️, and Siemens®️).
- Bilirubin (Total and Direct): Bilirubin levels were determined in plasma or serum samples using colorimetric methods, such as the Jendrassik-Grof method, on automated analyzers, ensuring high precision.
Regarding reference values, we acknowledge that while slight variations in reference ranges may exist across centers due to differences in equipment, local protocols, and target populations, all laboratories adhere to standardized methodologies as well as internal and external quality programs, ensuring consistency and comparability. To account for potential variability, we adjusted our analyses for GA rather than relying solely on fixed reference ranges. This approach allowed for a more nuanced comparison of the groups, focusing on statistically significant differences rather than predefined thresholds.
To improve organization and clarity, section 4.3 was renamed from "Demographic and Clinical Data" to "Demographic, Clinical, and Laboratory Data," reflecting its expanded content. These updates have been incorporated into the revised manuscript and can be found in lines 687-734.
If further detail about the equipment or methods is required, we would be happy to provide additional information to enhance the clarity of the manuscript.
Round 2
Reviewer 1 Report
Comments and Suggestions for Authors
The manuscript has been revised appropriately.
Author Response
Thank you for your comments
Reviewer 3 Report
Comments and Suggestions for Authors
The authors responded adequately to the comments from the previous review - I have no additional comments.
I congratulate the authors on their efforts and wish them continued success in their scientific career.
Author Response
Thank you for your comments